# Towards training digitally-tied analog blocks
# *via* hybrid gradient computation

**Timothy Nest**[*][♦]
timothy.nest@mila.quebec

**Maxence Ernoult**[†][♦]
maxence@rain.ai

## Abstract

Power efficiency is plateauing in the standard digital electronics realm such that new hardware, models, and algorithms are needed to reduce the costs of AI training. The combination of energy-based analog circuits and the Equilibrium Propagation (EP) algorithm constitutes a compelling alternative compute paradigm for gradient-based optimization of neural nets. Existing analog hardware accelerators, however, typically incorporate digital circuitry to sustain auxiliary non-weight-stationary operations, mitigate analog device imperfections, and leverage existing digital platforms. Such heterogeneous hardware lacks a supporting theoretical framework. In this work, we introduce *Feedforward-tied Energy-based Models* (ff-EBMs), a hybrid model comprised of feedforward and energy-based blocks housed on digital and analog circuits. We derive a novel algorithm to compute gradients end-to-end in ff-EBMs by backpropagating and "eq-propagating" through feedforward and energy-based parts respectively, enabling EP to be applied flexibly on realistic architectures. We experimentally demonstrate the effectiveness of this approach on ff-EBMs using Deep Hopfield Networks (DHNs) as energy-based blocks, and show that a standard DHN can be arbitrarily split into any uniform size while maintaining or improving performance with increases in simulation speed of up to four times. We then train ff-EBMs on ImageNet32 where we establish a new state-of-the-art performance for the EP literature (46 top-1 %) [3]. Our approach offers a principled, scalable, and incremental roadmap for the gradual integration of self-trainable analog computational primitives into existing digital accelerators.

## 1 Introduction

Gradient-based optimization, the cornerstone and most energy greedy component of deep learning, fundamentally relies upon three factors: i) highly parallel digital hardware such as GPUs, ii) feedforward models and iii) backprop (BP). With skyrocketing demands of AI compute, reducing the energy consumption of AI systems has become a matter of great economic, societal and environmental urgency [Strubell et al., 2020], calling for the exploration of novel compute paradigms [Thompson et al., 2020, Scellier, 2021, Stern and Murugan, 2023].

One promising path towards this goal is analog in-memory computing [Sebastian et al., 2020]: by mapping weights onto a crossbar of resistive devices, Kirchoff current and voltage laws inherently perform matrix-vector multiplications with constant time complexity [Cosemans et al., 2019]. By stacking multiple such crossbars, an entire neural network can be mapped onto a physical system. An important formalism for such a system is that of *energy-based* (EB) analog circuits

---

[*]Montreal Institute of Learning Algorithms (MILA)

[†]Rain AI

[♦]Equal contribution

[3]Our code is available on `https://github.com/rain-neuromorphics/hybrid_bp_ep_official`

38th Conference on Neural Information Processing Systems (NeurIPS 2024).

[Kendall et al., 2020, Stern et al., 2023, Dillavou et al., 2023, Scellier, 2024], which are "self-learning" systems that can compute loss gradients through two relaxations to equilibrium (i.e. two "forward passes"). Such a procedure falls under the umbrella of energy-based learning (EBL) algorithms [Scellier et al., 2024]. One such algorithm, Equilibrium Propagation (EP) [Scellier and Bengio, 2017], particularly stands out for its strong theoretical guarantees, relative scalability in the realm of backprop alternatives [Laborieux and Zenke, 2022, 2023] and proven application on small analog systems with $10,000\times$ greater energy-efficiency and substantial speedups compared to its GPU-based counterpart [Yi et al., 2023]. This suggests a new alternative compute paradigm for gradient-based optimization consisting of: i) analog hardware, ii) EBMs, and iii) EP.

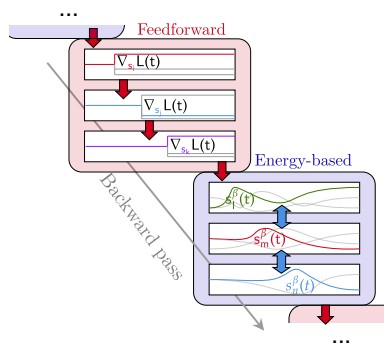

Figure 1: Illustrating BP-EP backward gradient chaining through feedforward (red) and energy-based (blue) blocks, accounting for digital and analog circuits respectively.

In this paper, we propose a theoretical framework for extending end-to-end gradient computation to a realistic setting where the system in question may or may not be *fully* analog. Such a setting is plausible in the near term, for two major reasons. First, analog circuits exhibit non-ideal physical behaviors which affect both the inference pathway [Wang et al., 2023, Ambrogio et al., 2023] and parameter optimization [Nandakumar et al., 2020, Spoon et al., 2021, Lammie et al., 2024], compromising performance. Second, owing to the latency and energy-consumption of resistive devices' write operations, analog circuits should be fully weight stationary – weights must be written before the inference procedure begins – which excludes many operations used conventionally in machine learning including activation functions, normalization, and attention [Spoon et al., 2021, Jain et al., 2022, Liu et al., 2023, Li et al., 2023]. Therefore, analog systems are likely to be used in combination with auxiliary digital circuitry, resulting in hybrid mixed precision systems [Haensch et al., 2018]. While the design of purely inferential engines made up of analog and digital parts is nearing commercial maturity [Ambrogio et al., 2023], *in-situ* learning of such systems has barely been explored. An important challenge remains in proving EBL algorithms can scale in a manner comparable to backprop, given the requirement of *simulating* EB systems on GPUs. Because of the necessity of convergence, this amounts in practice in performing lengthy root finding algorithms to simulate physical equilibrium, limiting proof-of-concepts thereof to relatively shallow (5-6 layer) models [Scellier et al., 2024, Scellier, 2024].

Our work contends that the best of both worlds can be achieved with the following triad: i) hybrid digital *and* analog hardware, ii) feedforward *and* EB models, iii) BP *and* EP. Namely, by modeling digital and analog parts as feedforward and EB modules respectively, we show how backprop and EP error signals can be chained end-to-end via feedforward and EB blocks respectively in a principled fashion. Rather than opposing digital and analog, or backprop and "alternative" learning algorithms, as is often done in the literature, we propose a novel hardware-aware building block which can, in principle, leverage advances from *both* digital and analog hardware in the near-term. More specifically:

• We propose *Feedforward-tied Energy-based Models* (ff-EBMs, Section 3.1) as high-level models of mixed precision systems whose inference pathway read as the composition of feedforward and EB modules (Eq. (5), Alg. 1).

• We show that gradients in ff-EBMs can be computed in an end-to-end fashion (Section 3.3), backpropagating through feedforward blocks and "eq-propagating" through EB blocks (Theorem 3.1, Alg. 2) and that this procedure is rooted in a deeply-nested optimization problem (Section 3.2).

• Finally, we experimentally demonstrate the effectiveness of our algorithm on ff-EBMs where EBM blocks are Deep Hopfield Networks (DHNs) (Section 4).In particular we show that i) final and *transient* gradient estimates computed by our algorithm (Alg. 2) near perfectly match gradients computed by end-to-end automatic differentiation (Section 4.2), which we also prove mathematically (Theorem 4.1), ii) a standard DHN model can be arbitrarily split into a ff-DHN with the equivalent

layers and architectural layers while maintaining or improving performance, remaining on par with automatic differentiation and being *up to four times faster to simulate* depending on the convergence criterion at use to compute equilibrium (Section 4.3), iii) the proposed approach yields 46 % top-1 (70% top-5) validation accuracy on ImageNet32 when training a ff-EBM of 15 layers, beating current state-of-the-art for EP by a large margin, without relying on holomorphic transformations inside EBM blocks [Laborieux and Zenke, 2022, 2023]

## 2 Background

**Notations.** Given $A : \mathbb{R}^n \to \mathbb{R}^m$ a differentiable mapping, we denote its *total* derivative with respect to $s_j$ as $d_{s_j}A(s) := dA(s)/ds_j \in \mathbb{R}^m$, its *partial* derivative with respect to $s_j$ as $\partial_j A(s) := \partial A(s)/\partial s_j \in \mathbb{R}^m$. When $A$ takes scalar values ($m = 1$), its *gradient* with respect to $s_j$ is denoted as $\nabla_j A(s) := \partial_j A(s)^\top$.

### 2.1 Energy-based models (EBMs)

For a given static input and set of weights, Energy-based models (EBMs) implicitly yield a prediction through the minimization of an energy function. As such they are a particular kind of implicit model. Namely, an EBM is defined by a (scalar) energy function $E : s, \theta, x \to E(s, \theta, x) \in \mathbb{R}$ where $x$, $s$, and $\theta$ respectively denote a static input, hidden and output neurons and model parameters, and each such tuple defines a configuration with an associated scalar energy value. Among all configurations for a given input $x$ and some model parameters $\theta$, the model prediction $s_\star$ is implicitly given as an equilibrium state which minimizes the energy function:

$$s_\star := \arg \min_s E(s, \theta, x). \tag{1}$$

### 2.2 Standard bilevel optimization

Assuming that $\nabla_s^2 E(x, s_\star, \theta)$ is invertible, note that the equilibrium state $s_\star$ implicitly depends on $x$ and $\theta$ by virtue of the implicit function theorem [Dontchev et al., 2009]. Therefore our goal when training an EBM–in a supervised setting, for instance – is to adjust the model parameters $\theta$ such that $s_\star(x, \theta)$ minimizes some cost function $\ell : s, y \to \ell(s, y) \in \mathbb{R}$ where $y$ is some ground-truth label associated to $x$. More formally, this learning objective can be stated with the following *bilevel optimization problem* [Zucchet and Sacramento, 2022]:

$$\min_\theta \mathcal{C}(x, \theta, y) := \ell(s_\star, y) \quad \text{s.t.} \quad s_\star = \arg \min_s E(s, \theta, x) \tag{2}$$

Solving Eq. (2) in practice amounts to computing the gradient of its outer objective $\mathcal{C}(x, \theta)$ with respect to $\theta$ ($d_\theta \mathcal{C}(x, \theta)$) and then performing gradient descent over $\theta$.

### 2.3 Equilibrium Propagation (EP)

An algorithm used to train an EBM model in the sense of Eq. (2) may be called an EBL algorithm [Scellier et al., 2024]. Equilibrium Propagation (EP) [Scellier and Bengio, 2017] is an EBL algorithm which computes an estimate of $d_\theta \mathcal{C}(x, \theta)$ with at least two phases. During the first phase, the model is allowed to evolve freely to $s_\star = \arg \min_s E(s, \theta, x)$. Then, the model is slightly nudged towards decreasing values of cost $\ell$ and settles to a second equilibrium state $s_\beta$. This amounts to augmenting the energy function $E$ by an additional term $\beta \ell(s, y)$ where $\beta \in \mathbb{R}^\star$ is called the *nudging factor*. Next, the weights are updated to increase the energy of $s_\star$ and decrease that of $s_\beta$, thereby "contrasting" these two states. More formally, Scellier and Bengio [2017] show in the seminal EP paper:

$$s_\beta := \arg \min_s \left[E(s, \theta, x) + \beta \ell(s, y)\right], \quad \Delta \theta^{\mathrm{EP}} := \frac{\alpha}{\beta} \left(\nabla_2 E(s_\star, \theta, x) - \nabla_2 E(s_\beta, \theta, x)\right), \tag{3}$$

where $\alpha$ denotes some learning rate. EP comes in different flavors depending on the sign of $\beta$ inside Eq. (3) or on whether two nudged states of opposite nudging strengths ($\pm \beta$) are contrasted, a variant called *Centered* EP (C-EP) which was shown to work best in practice [Laborieux et al., 2021, Scellier et al., 2024] and reads as:

$$\Delta \theta^{\mathrm{C-EP}} := \frac{\alpha}{2\beta} \left(\nabla_2 E(s_{-\beta}, \theta, x) - \nabla_2 E(s_\beta, \theta, x)\right), \tag{4}$$

# 3 Tying energy-based models with feedforward blocks

In the present section we introduce a new model, *Feedforward-tied EBMs* (ff-EBMs, section 3.1), which read as composition of feedforward and EB transformations (Alg. 1). We show how optimizing ff-EBMs amounts to solving a multi-level optimization problem (Section 3.2) and propose a BP-EP gradient chaining algorithm as a solution(Section 3.3, Theorem 3.1, Alg. 2). We highlight as an edge case the fact that ff-EBMs reduce to standard feedforward nets (Lemma A.1) and the proposed BP-EP gradient chaining algorithm to standard BP (Corollary A.1) when each EB block comprises a single hidden layer. We highlight in red and blue the parts of the model and associated algorithms performed inside feedforward (digital) and EB (analog) blocks respectively.

## 3.1 Feedforward-tied Energy-based Models (ff-EBMs)

**Inference procedure.** We define *Feedforward-tied Energy-based Models* (ff-EBMs) as compositions of feedforward and EB transformations. Namely, an data sample $x$ is fed into the first feedforward transformation $F^1$ parametrized by some weights $\omega^1$, which yields an output $x_\star^1$. Then, $x_\star^1$ is fed as a static input into the first EB block $E^1$ with parameters $\theta^1$, which relaxes to an equilibrium state $s_\star^1$. $s_\star^1$ is in turn fed into the next feedforward transformation $F^1$ with weights $\omega^1$ and the above procedure repeats until reaching the output layer $\hat{o}$. More formally, denoting $F^k$ and $E^k$ the $k^{\text{th}}$ feedforward and EB blocks parametrized by the weights $\omega^k$ and $\theta^k$ respectively, the inference pathway of a ff-EBM reads as:

$$\begin{cases} s^0 := x \\ x_\star^k := F^k(s_\star^{k-1}, \omega^k), \quad s_\star^k := \arg\min_s E^k(s, \theta^k, x_\star^k) \quad \forall k = 1 \cdots N-1 \\ \hat{o}_\star := F^N(s_\star^{N-1}, \omega^N) \end{cases} \quad (5)$$

The ff-EBM inference procedure is depicted more compactly inside Fig. 2 (left) and Alg. 1.

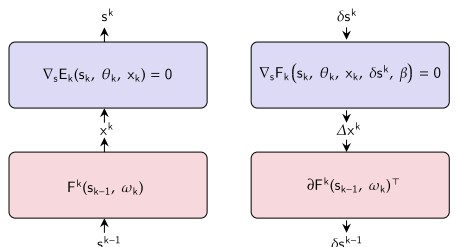

**Algorithm 1** ff-EBM inference (Eq. (5))

1: $s \leftarrow x$
2: **for** $k = 1 \cdots N - 1$ **do**
3: $\quad x \leftarrow F^k(s, \omega^k)$
4: $\quad s \leftarrow \underset{s}{\mathbf{Optim}}\left[E^k(s, \theta^k, x)\right]$
5: **end for**
6: $\hat{o} \leftarrow F^N(s, \omega^N)$

Figure 2: Depiction of the forward (left) and backward (right) pathways through a ff-EBM, with blue and pink blocks denoting EB and feedforward transformations.

**Form of the energy functions.** We further specify the form of the energy of the $k^{\text{th}}$ EB block of a ff-EBM as defined per Eq. (5). The associated energy function $E^k$ takes some static input $x^k$ from the output of the preceding feedforward transformation, has hidden neurons $s^k$ and is parametrized by weights $\theta^k$. More precisely:

$$E^k(s^k, \theta^k, x^k) := G^k(s^k) - s^{k^\top} \cdot x^k + U^k(s^k, \theta^k) \quad (6)$$

Eq. (6) reveals three different contributions to the energy. The first term determines the non-linearity applied inside the EB block [Zhang and Brand, 2017, Høier et al., 2023]: for a given invertible and continuous activation function $\sigma$, $G$ is defined such that $\nabla G = \sigma^{-1}$ (see Appendix A.1.3).

The second term inside Eq. (6) accounts for a purely feedforward contribution from the previous feedforward block $F^k$. Finally, the third term accounts for *internal* interactions within the layers of the EB block.

**Recovering a feedforward net.** When taking the gradient of $E^k$ as defined in Eq. (6) with respect to $s^k$ and zeroing it out, it can be seen that $s_\star^k$ is implicitly defined as:

$$s_\star^k := \sigma\left(x^k - \nabla_1 U^k(s_\star^k, \theta^k)\right) \quad (7)$$

A noteworthy edge case highlighted by Eq. (7) is when $U^k = 0$ for all $k$'s, i.e. when there are no intra-block layer interactions, or equivalently when the EB block comprises a single layer only. In this case, $s_\star^k$ is simply a feedforward mapping $x^k$ through $\sigma$ and in turn the ff-EBM is simply a standard feedforward architecture (see Lemma A.1 inside Appendix A.1.1).

## 3.2 Multi-level optimization of ff-EBMs

Just as learning EBMs can be naturally cast as a bilevel optimization problem, learning ff-EBMs equates to a *multi-level* optimization problem where the variables being optimized in the inner subproblems are comprised of EB block variables $s^1, \cdots, s^{N-1}$. To make this clearer, we re-write the energy function of the $k^{\text{th}}$ block $E^k$ from Eq. (6) to highlight the dependence between two consecutive EB block states:

$$\widetilde{E}^k(s^k, \theta^k, s_\star^{k-1}, \omega^k) := E^k\left(s^k, \theta^k, F^k\left(s_\star^{k-1}, \omega^{k-1}\right)\right) \tag{8}$$

It can be seen from Eq. (8) that the equilibrium state $s_\star^k$ obtained by minimizing $E^k$ will be dependent upon the equilibrium state $s_\star^{k-1}$ of the previous EB block, which propagates back through prior EB blocks. Denoting $W := \{\theta^1, \cdots, \theta^{N-1}, \omega^1, \cdots, \omega^N\}$, the learning problem for a ff-EBM can therefore be written as:

$$\min_W \mathcal{C}(x, W, y) := \ell(\hat{o}_\star = F^N(s_\star^{N-1}, \omega^N), y) \tag{9}$$

$$\text{s.t.} \quad s_\star^{N-1} = \arg\min_s \widetilde{E}^{N-1}(s, \theta^{N-1}, s_\star^{N-2}, \omega^{N-1}) \quad \cdots \quad \text{s.t.} \quad s_\star^1 = \arg\min_s \widetilde{E}^1(s, \theta^1, x, \omega^1)$$

Here again and similarly to bilevel optimization, solving Eq. (9) in practice amounts to computing $g_{\theta^k} := d_{\theta^k}\mathcal{C}$ and $g_{\omega^k} := d_{\omega^k}\mathcal{C}$ and performing gradient descent on $\theta^k$ and $\omega^k$.

## 3.3 A BP–EP gradient chaining algorithm

**Main result: explicit BP-EP chaining.** Based on the multilevel optimization formulation of ff-EBMs learning in Eq. (9), we state the main theoretical result of this paper in Theorem 3.1 (see proof in Appendix A.2.1).

**Theorem 3.1** (Informal). *Assuming a model of the form Eq. (5), we denote $s_\star^1, x_\star^1, \cdots, s_\star^{N-1}, \hat{o}_\star$ the states computed during the forward pass as depicted in Alg. 1. We define the nudged state of block $k$, denoted as $s_\beta^k$, implicitly through $\nabla_1 \mathcal{F}^k(s_\beta^k, \theta^k, x_\star^k, \delta s^k, \beta) = 0$ with:*

$$\mathcal{F}^k(s^k, \theta^k, x_\star^k, \delta s^k, \beta) := E^k(s^k, \theta^k, x_\star^k) + \beta s^{k^\top} \cdot \delta s^k \tag{10}$$

*Denoting $\delta s^k$ and $\Delta x^k$ the error signals computed at the input of the feedforward block $F^k$ and of the EB block $E^k$ respectively, then the following chain rule applies:*

$$\delta s^{N-1} := \nabla_{s^{N-1}}\ell(\hat{o}_\star, y), \quad g_{\omega^N} = \nabla_{\omega^N}\ell(\hat{o}_\star, y) \tag{11}$$

$$\forall k = 2 \cdots N - 1:$$

$$\begin{cases} \Delta x^k = d_\beta\left(\nabla_3 E^k(s_\beta^k, \theta^k, x_\star^k)\right)\Big|_{\beta=0}, \quad g_{\theta^k} = d_\beta\left(\nabla_2 E^k(s_\beta^k, \theta^k, x_\star^k)\right)\Big|_{\beta=0} \\ \delta s^{k-1} = \partial_1 F^k\left(s_\star^{k-1}, \omega^k\right)^\top \cdot \Delta x^k, \quad g_{\omega^k} = \partial_2 F^k\left(s_\star^{k-1}, \omega^k\right)^\top \cdot \Delta x^k \end{cases} \tag{12}$$

**Proposed algorithm: implicit BP-EP chaining.** Theorem 3.1 reads intuitively. It prescribes an *explicit* chaining of EP error signals passing backward through $E^k$ ($\delta s^k \to \Delta x^k$) and BP error signals passing backward through $\partial F^{k^\top}$ ($\Delta x^k \to \delta s^{k-1}$), which directly mirrors the ff-EBM inference pathway as depicted in Fig. 2. Yet noticing that:

$$\begin{cases} \delta s^{k-1} = \partial_1 F^k\left(s_\star^{k-1}, \omega^k\right)^\top \cdot \Delta x^k = d_\beta\left(\nabla_3 \widetilde{E}^k\left(s_\beta^k, \theta^k, s_\star^{k-1}, \omega^k\right)\right)\Big|_{\beta=0}, \\ g_{\omega^k} = \partial_2 F^k\left(s_\star^{k-1}, \omega^k\right)^\top \cdot \Delta x^k = d_\beta\left(\nabla_4 \widetilde{E}^k\left(s_\beta^k, \theta^k, s_\star^{k-1}, \omega^k\right)\right)\Big|_{\beta=0}, \end{cases}$$

the same error signal can by passed through $\widetilde{E}^k$ ($\delta s^k \to \delta s^{k-1}$) where BP and EP are *implicitly* chained inside $\widetilde{E}^k$ (see Appendix A.2.1). This insight, along with a centered scheme to estimate

derivatives with respect to $\beta$ around 0 as done for the C-EP algorithm (Eq. (4)), motivates the implicit BP-EP gradient chaining algorithm in Alg. 2 we used for our experiments (see Alg. 5 inside Appendix A.3.1 for its explicit counterpart). Given that the proposed algorithm appears as a a generalization of EP, we refer to Alg. 2as "EP" in the experimental section, for simplicity.

---

**Algorithm 2** Implicit BP-EP gradient chaining (Theorem (3.1))

---

1: $\delta s, g_{\omega^N} \leftarrow \nabla_{s^{N-1}}\ell(\hat{o}_\star, y), \nabla_{\omega^N}\ell(\hat{o}_\star, y)$          ▷ Single BP step

2: **for** $k = N - 1 \cdots 1$ **do**

3:    $s_\beta \leftarrow \underset{s}{\textbf{Optim}}\left[\widetilde{E}^k(s, \theta^k, s_\star^{k-1}, \omega^k) + \beta s^\top \cdot \delta s\right]$       ▷ EP in $\widetilde{E}^k$

4:    $s_{-\beta} \leftarrow \underset{s}{\textbf{Optim}}\left[\widetilde{E}^k(s, \theta^k, s_\star^{k-1}, \omega^k) - \beta s^\top \cdot \delta s\right]$

5:    $g_{\theta^k} \leftarrow \frac{1}{2\beta}\left(\nabla_2\widetilde{E}^k(s_\beta, \theta^k, s_\star^{k-1}, \omega^k) - \nabla_2\widetilde{E}^k(s_{-\beta}, \theta^k, s_\star^{k-1}, \omega^k)\right)$

6:    $g_{\omega^k} \leftarrow \frac{1}{2\beta}\left(\nabla_4\widetilde{E}^k(s_\beta, \theta^k, s_\star^{k-1}, \omega^k) - \nabla_4\widetilde{E}^k(s_{-\beta}, \theta^k, s_\star^{k-1}, \omega^k)\right)$    ▷ Implicit BP in $F^k$

7:    $\delta s \leftarrow \frac{1}{2\beta}\left(\nabla_3\widetilde{E}^k(s_\beta, \theta^k, s_\star^{k-1}, \omega^k) - \nabla_3\widetilde{E}^k(s_{-\beta}, \theta^k, s_\star^{k-1}, \omega^k)\right)$

8: **end for**

---

**Recovering backprop.** When the ff-EBM under consideration is purely feedforward ($U^k = 0$), we show that Eqs. (11)–(12) reduce to standard BP through a feedforward net (Corollary A.1, Alg. 6 and Alg. 7 in Appendix A.2.1). Since this case is extremely close to standard BP through feedforward nets, we do not consider this setting in our experiments.

## 4 Experiments

In this section, we present the ff-EBMs used in our experiments (Section 4.1) and carry out *static* gradient analysis – computing and analyzing ff-EBM parameter gradients for some $x$ and $y$ (Section 4.2). We extend the observation made by Ernoult et al. [2019] –that *transient* EP parameter gradients obtained during the second phase match those computed by automatic differentiation through equilibrium and across blocks– to ff-EBMs (Fig. (3)–(4), Theorem 4.1), showing that gradient estimates of automatic differentiation and EP in our framework, are near perfectly aligned (Fig. 5). We then show on the CIFAR-10 task that performance of ff-EBMs can be maintained or improved across various block splits maintaining the same number of layers, while remaining on par with automatic differentiation(Section 4.3). We show furthermore that blocks of smaller size are up to four times faster to simulate depending on the convergence criterion at use for computing equilibrium inside EB blocks. Finally, we perform further ff-EBM training experiments on CIFAR-100 and ImageNet32 where we establish a new state-of-the-art performance in the EP literature (Section 4.4).

### 4.1 Setup

**Models.** Using the same notations as in Eq. (6), the ff-EBMs at use in this section are defined:

$$\begin{cases} U_{\text{FC}}^k(s^k, \theta^k) := -\frac{1}{2}s^{k\top} \cdot \theta^k \cdot s^k, \\ U_{\text{CONV}}^k(s^k, \theta^k) := -\frac{1}{2}s^k \bullet (\theta^k \star s^k) \end{cases} , \begin{cases} F_{\text{BN}}^k(s^{k-1}, \omega^k) := \text{BN}\left(\mathcal{P}\left(\omega_{\text{CONV}}^k \star s_L^{k-1}\right); \omega_\alpha^k, \omega_\beta^k\right), \\ F_{\text{ID}}^k(s^{k-1}) := s_L^{k-1} \end{cases}$$

(13)

with $\text{BN}(\cdot; \omega_\alpha^k, \omega_\beta^k)$, $\mathcal{P}$ and $\star$ the batchnorm, pooling and convolution operations, $\bullet$ the generalized dot product for tensors and $s^k := \left(s_1^{k\top}, \cdots s_L^{k\top}\right)^\top$ the state of block $k$ comprising $L$ layers. Such EBM blocks are known as Deep Hopfield Networks (DHNs). DHNs are comprised of fully connected ($U_{\text{FC}}^k$) and convolutional operations ($U_{\text{CONV}}^k$) forming a symmetric weight matrix $\theta^k$ with a sparse, block-wise structure such that each layer $s_\ell^k$ is bidirectionally connected to its neighboring layers $s_{\ell-1}^k$ and $s_{\ell+1}^k$ through connections $\theta_{\ell-1}^k$ and $\theta_\ell^{k\top}$ respectively (see Appendix A.1.3). To empirically ensure convergence, the non-linearity $\sigma$ applied within EB blocks is $\sigma_\alpha(x) := \min\left(\max\left(\alpha x, 0\right), 1\right)$ with $\alpha \in (0, 1)$. Finally, two design choices were *instrumental* to the success of ff-EBM gradient computation and subsequent training. First, we initialized the weights of $U_{\text{FC}}^k$ and $U_{\text{CONV}}^k$ using

*Gaussian Orthogonal Ensembles* (GOE) [Agarwala and Schoenholz, 2022] to enable faster equilibrium computation (see next paragraph). Second, while the last layer of a given block was simply passed as an input to the next block (i.e. using $F_{\text{ID}}^k$ in Eq. (13)) for small enough models ($L = 6$ inside the experiment depicted in Section 4.3), the use of batchnorm layers in between blocks (i.e. using $F_{\text{BN}}^k$ in Eq. (13)) becomes essential for deeper models.

**Equilibrium computation.**   As depicted in Alg. 2, the steady states $s_{\pm\beta}$ may be computed with any loss minimization algorithm. Here, as in past works on EP [Ernoult et al., 2019, Laborieux et al., 2021, Laborieux and Zenke, 2022, Scellier et al., 2024], we employ a fixed-point iteration scheme to compute the EB blocks steady states. Namely, we iterate Eq. (7) until reaching equilibrium (the same scheme is used for ff-EBM inference, Alg. 1, with $\beta = 0$.):

$$s_{\pm\beta,t+1}^k \leftarrow \sigma \left( x^k - \nabla_1 U^k(s_{\pm\beta,t}^k, \theta^k) \mp \beta \delta s^k \right) \qquad (14)$$

Important details about how Eq. (14) is executed in practice have to be highlighted. First, we employ a scheme to *asynchronously* update even ($s_{2\ell'}^k$) and odd ($s_{2\ell'+1}^k$) layers [Scellier et al., 2024] – see Appendix A.1.3. Second, Eq. (14) were either executed for a *fixed* and predetermined number of steps as done in the aforementioned EP literature, or using an $\epsilon$−tolerance-based convergence criterion (TOL) which stops executing Eq. (14) when $(s_{t+1}^i - s_t^i)/s_t^i \leq \epsilon$ *on average* – see Appendix A.5.3 for details.

**Algorithm baseline.**   As an algorithmic baseline, we simply use automatic differentiation (AD) backward through the fixed-point iteration scheme Eq. (14) with $\beta = 0$ and directly initializing $s_{t=0}^k = s_\star$ (Fig. 4). This version of AD, where we *backpropagate through equilibrium*, is known as "Recurrent Backpropagation" [Almeida, 1987, Pineda, 1987] or Implicit Differentiation (ID).

### 4.2   Static comparison of EP and ID on ff-EBMs

In order to study the *transient dynamics* of ID and EP, we define, with $W^k := \{\theta^k, \omega^k\}$:

$$\begin{cases} \widehat{g}_{W^k}^{\text{ID}}(t) := \sum_{k=0}^T d_{W^k(T-k)} \mathcal{C}(x, W^k, y), \\ \widehat{g}_{W^k}^{\text{EP}}(\beta, t) := \frac{1}{2\beta} \left( \nabla_{W^k} \widetilde{E}^k(s_{\beta,t}^k, W^k, s_\star^{k-1}) - \nabla_{W^k} \widetilde{E}^k(s_{-\beta,t}^k, W^k, s_\star^{k-1}) \right), \end{cases} \qquad (15)$$

where $s_{\pm\beta,t}^k$ is computed from Eq. (14) with the nudging error current $\delta s^k$ computed with Alg. 2, and $T$ is the total number of iterations used for both ID and EP in the gradient computation phase.

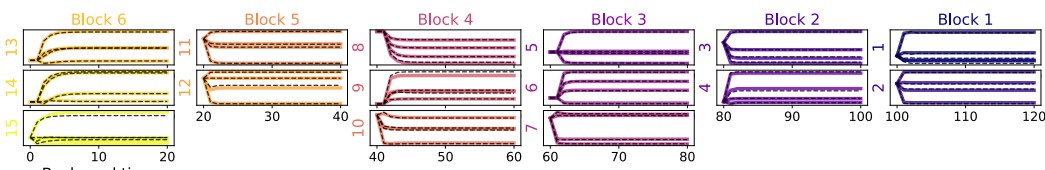

Figure 3: EP and ID partially computed gradients $((\widehat{g}_w^{\text{EP}}(t))_{t\geq0}$ in black dotted curves and $(\widehat{g}_w^{\text{ID}}(t))_{t\geq0}$ in plain colored curves) going *backward through equilibrium* for ID and *forward through the nudging phase* for EP [Ernoult et al., 2019] for a random sample $x$ and associated label $y$. The ff-EBM comprises 6 blocks and 15 layers in total, with block sizes of either 2 or 3 layers. Each sub-panel represents a layer (labeled on the y-axis) with each curve corresponding to a randomly selected weight. "Backward" time is indexed from $t = 0$ to $T = 120$, starting from block 6 backward to block 1, with 20 fixed-point iteration dynamics (Eq. (14)) being used for both EP and ID within each block.

For a given block $k$, $d_{W^k(T-k)}\mathcal{C}(x, W, y)$ is the "sensitivity" of the loss $\mathcal{C}$ to parameter $W^k$ at timestep $T - k$ so that $\widehat{g}_{W^k}^{\text{ID}}(t)$ is a ID gradient *truncated* at $T - t$. Fig. 4 depicts the computational graph that is differentiated through when using ID and shows where $\widehat{g}_{W^k}^{\text{ID}}(t)$ are obtained correspondingly. Similarly, $\widehat{g}_{W^k}^{\text{EP}}(t)$ is an EP gradient truncated at $t$ steps forward through the nudged phase. When $T$ is sufficiently large, $\widehat{g}_{W^k}^{\text{ID}}(T)$ and $\widehat{g}_{W^k}^{\text{EP}}(T)$ converge to $d_{W^k}\mathcal{C}(x, W, y)$. Fig. 3 displays $(\widehat{g}_{W^k}^{\text{ID}}(t))_{t\geq0}$ and $(\widehat{g}_{W^k}^{\text{EP}}(t))_{t\geq0}$ on an heterogeneous ff-EBM of 6 blocks and 15 layers (16 if counting the last linear

"readout" layer computing the logits) with blocks comprising 2 or 3 layers for a randomly selected sample $x$ and its associated label $y$ – see caption for a detailed description. It can be seen EP and ID error weight gradients qualitatively match very well throughout time, across layers and blocks. We also display the cosine similarity between the final EP and ID weight gradient estimate $\widehat{g}^{\mathrm{ID}}_{W^k}(T)$ and $\widehat{g}^{\mathrm{EP}}_{W^k}(T)$ for each layer and observe that EP and ID weight gradients are near perfectly aligned. Theorem 4.1 generalizes the equivalence between EP and ID to ff-EBMs [Ernoult et al., 2019].

**Theorem 4.1** (Informal). *Assuming $\forall k = 1 \cdots N - 1:\ s_0^k = \cdots = s_\tau^k = s_\star^k$:*

$$\forall k = 1 \cdots N - 1,\ \forall t = 0 \cdots \tau:\quad \hat{g}^{\mathrm{AD}}_{W^k}(t) = \hat{g}^{\mathrm{ID}}_{W^k}(t) = \lim_{\beta \to 0} \hat{g}^{\mathrm{EP}}_{W^k}(\beta, t) \tag{16}$$

Figure 4: **Light grey:** computational graph associated with ff-EBM inference (Alg. 1) when applying fixed-point iteration to compute equilibrium states within each block (Eq. (14)) where the node $s_t^k$ denotes the state of block $k$ (comprising several layers) at timestep $t$. **Blue arrows:** backward automatic differentiation (AD) through the computational graph where $\hat{g}^{\mathrm{ID}}_{W^k}(t)$ is the partially computed gradient truncated at $T - t$. Since the states which are differentiated through are taken at *equilibrium* ($s_t^k = s_\star^k\ \forall t = 0 \cdots \tau$) this instantiation of AD can be viewed as *Implicit Differentiation* (ID).

## 4.3 Splitting experiment

For a given (standard, single block) EBM with a *fixed* number of layers, we ask how block splitting of this EBM into a ff-EBM with multiple EB blocks affects training performance and Wall Clock (simulation) Time (WCT). We address this question with two different depths ($L = 6$ and $L = 12$ layers in total) and various block sizes (bs), maintaining a fixed total number of layers (e.g. for $L = 6$, 1 block of 6 layers, 2 blocks of 3 layers, etc.). Additionally, to ensure the fairest comparison in terms of WCTs across different splits, *we adopt the aforementioned TOL approach to execute the fixed-point dynamics Eq. (14) within each EB block.* We display the results obtained on the CIFAR-10 task inside Table 1. We observe that EP performance improves with smaller block sizes

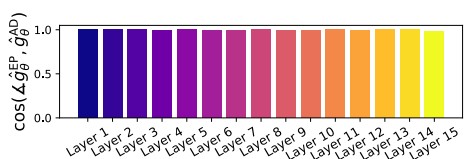

Figure 5: Cosine similarity between EP and ID weight gradients on a randomly selected sample $x$ and associated label $y$ in the same setting as Fig. 3 using the same color code to label the layers. We observe near-perfect alignment between EP and ID gradients. See Fig. 7 for a precise depiction of the model at use.

Table 1: Validation accuracy and Wall Clock Time (WCT) obtained on CIFAR-10 by EP (Alg. 2) and ID on models with different number of layers ($L$) and block sizes ("bs"). 3 seeds are used.

|  | EP | | ID | |
|---|---|---|---|---|
|  | Top-1 (%) | WCT | Top-1 (%) | WCT |
| **L =6** | | | | |
| bs=6 | $89.2^{\ \pm 0.2}$ | **7:01** | $87.3^{\ \pm 0.4}$ | **6:51** |
| bs=3 | $89.8^{\ \pm 0.2}$ | **5:17** | $89.3^{\ \pm 0.2}$ | **5:10** |
| bs=2 | $90.1^{\ \pm 0.1}$ | **3:57** | $90.0^{\ \pm 0.1}$ | **4:05** |
| **L =12** | | | | |
| bs=4 | $89.4^{\ \pm 0.7}$ | **11:59** | $89.5^{\ \pm 0.2}$ | **8:28** |
| bs=3 | $\mathbf{92.5}^{\ \pm 0.1}$ | **7:33** | $92.0^{\ \pm 0.1}$ | **4:16** |
| bs=2 | $92.0^{\ \pm 0.2}$ | **3:14** | $91.5^{\ \pm 0.2}$ | **3:07** |

(reaching 90.1% and 92.5% for $L = 6$ and $L = 12$ respectively with bs $= 2$ and bs $= 3$) with overall WCT reduction (up to $\approx \times 4$) while remaining on par with ID. This significant reduction in

WCT is due to the fact that inference time for ff-EBMs with DHN blocks by construction scales *linearly* with the number of blocks rather than supralinearly with the number of layers as has been empirically observed in the EP literature [Ernoult et al., 2019]. When instead using a *fixed number of iterations* to execute Eq. (14) inside EB blocks (Table 5 in Appendix A.5.3), EP performance is *maintained* across all splits (90.1% and 92.5% for $L = 6$ and $L = 12$ resp.) and is still on par with ID. However, there is no advantage in terms of WCTs in this case as the number of iterations is kept the same across all block splits and is much larger than necessary for smaller block sizes. Results for $L = 6$ are consistent with the existing literature and those for $L = 12$ surpass EP state-of-the-art on CIFAR-10 [Scellier et al., 2024, Laborieux and Zenke, 2022]. Overall these results suggest that: i) ff-EBM performance is agnostic to EB block sizes and are therefore flexible in design, ii) ff-EBMs are much faster to simulate that EBM counterparts of equivalent depth.

## 4.4 Scaling experiment

We now consider ff-EBMs of *fixed* block size 2, and *relatively small* number of iterations . We train two models of depth ($L = 12$ and $L = 15$) on CIFAR-100 and ImageNet32 by EP and ID and show the results obtained in Table 2. Here again we observe that EP matches ID performance on all models and tasks, ff-EBMs benefit from depth, and the performance obtained by training the 15-layer deep ff-EBM by EP exceeds state-of-the-art performance on ImageNet32 by around 10% top-1 validation accuracy [Laborieux and Zenke, 2022] and by around 5% the best performance reported on this benchmark among all backprop alternatives [Høier et al., 2023].

Table 2: Validation accuracy and Wall Clock Time (WCT) obtained on CIFAR100 and ImageNet32 by EP and Autodiff on models with different number of layers ($L$) and a block size of 2 (bs=2). 3 seeds are used. We compare our results against best published results on ImageNet32 by EP [Laborieux and Zenke, 2022] and against all backprop alternatives [Høier et al., 2023].

| | | EP | | | ID | | |
|---|---|---|---|---|---|---|---|
| | | Top-1 (%) | Top-5 (%) | WCT | Top-1 (%) | Top-5 (%) | WCT |
| CIFAR100 | L=12 | $69.3^{\pm0.2}$ | $89.9^{\pm0.5}$ | 4:33 | $69.2^{\pm0.1}$ | $90.0^{\pm0.2}$ | 4:16 |
| | L=15 | $71.2^{\pm0.2}$ | $90.2^{\pm1.2}$ | 2:54 | $71.1^{\pm0.3}$ | $90.9^{\pm0.1}$ | 2:44 |
| ImageNet32 | L=12 | $44.7^{\pm0.1}$ | $61{:}00^{\pm0.1}$ | 65:23 | $44.7^{\pm0.6}$ | $68.9^{\pm0.6}$ | 57:00 |
| | L=15 | $\mathbf{46.0}^{\pm0.1}$ | $\mathbf{70.0}^{\pm0.2}$ | 46:00 | $45.5^{\pm0.1}$ | $69.0^{\pm0.1}$ | 40:01 |
| Laborieux and Zenke [2022] | | 36.5 | 60.8 | – | – | – | – |
| Høier et al. [2023] | | 41.5 | 64.9 | – | – | – | – |

## 5 Discussion

**EP literature.** Ever since fixed-point iteration schemes were first proposed to facilitate EP experiments [Ernoult et al., 2019, Laborieux et al., 2021], there has been a growing body of work assessing scalability of EP and its algorithmic extensions on standard vision tasks. Most notably, Laborieux and Zenke [2022] introduced a holomorphic version of EP where loss gradients are computed with adiabatic oscillations of the model by nudging in the complex plane, which was very recently extended to more general implicit models [Laborieux and Zenke, 2023]. Moving further towards physical implementations of EP, Scellier et al. [2022] proposed a fully black-box version of EP where details about the system may not be known. All these advances could be readily applied inside our EP-BP chaining algorithm to EB blocks. The work closest to ours, albeit with a purely theoretical motivation and without clear algorithmic prescriptions, is that of Zach [2021] where feedforward model learning is cast as a deeply nested optimization in which consecutive layers are tied by elemental pair-wise energy functions. This work more recently inspired the Dual Propagation algorithm [Høier et al., 2023]. Such a setting can be construed as a particular case of ff-EBM learning by EP where each EB block comprises a *single* layer ($U^k = 0$) inside Eq. (6)–which, as we have shown, is tantamount to standard BP(see last paragraph of Section 3.3).

**Forward-only learning beyond EP.** Given that *zeroth-order* (ZO) optimization and "forward-forward" (FF) algorithms [Dellaferrera and Kreiman, 2022, Hinton, 2022] can be applied to *any*

model, and–like EP– compute a learning rule through multiple inference steps, one may wonder why it is important that our models should be energy-based. While mechanistically appealing for analog hardware [Oguz et al., 2023, Momeni et al., 2023, 2024, Xue et al., 2024], these forward-only approaches do not match the performance of automatic differentiation on equivalent models, even if they are roughly the same size as those studied in our work. On the one hand, weight perturbation [Fiete et al., 2007] (WP or "SPSA" [Spall, 1998]), yields unbiased yet noisy gradient estimates with variance scaling cubically with the model dimensionality [Ren et al., 2022], resulting in a significant gap in model performance compared to backprop, that can only be partially mitigated when using heuristics [Silver et al., 2021, Ren et al., 2022, Fournier et al., 2023, Chen et al., 2023]. On the other hand FF algorithms, as learning heuristics, suffer from a lack of theoretical guarantees which may impact the resulting model performance.

**Limitations and future work.** Since our recipe advocates EP–BP chaining by construction, it is fair to say that ff-EBM learning partially inherits the pitfalls of BP. Fortunately, nothing prevents feedforward modules inside ff-EBMs from being trained by *any* BP alternative to mitigate specific issues. For instance: BP can be parameterized by feedback weights to obviate weight transport from the inference circuit to the gradient computation circuit [Akrout et al., 2019]; BP gradients can be approximated as finite differences of feedback operators [Ernoult et al., 2022]; or computed via implicit forward-mode differentiation [Hiratani et al., 2022, Fournier et al., 2023, Malladi et al., 2023]; local layer-wise self-supervised or supervised loss functions can be used to prevent "backward locking" [Belilovsky et al., 2019, Ren et al., 2022, Hinton, 2022]. This insight may help exploring many variants of ff-EBM training.

Pursuing the core motivation of this work, one natural extension of this study is to incorporate *more hardware realism into ff-EBMs*. Beyond Deep Hopfield networks, Deep Resistive Nets (DRNs) – developed by Scellier [2024] and strongly inspired by Kendall et al. [2020] – are exact models of idealized analog circuits, trainable by EP, promising fast simulation times. As such, using DRNs as EB blocks inside ff-EBMs is an exciting research direction – see Fig. 6. Still, further work in this direction presents new challenges especially given device non-idealities which may affect the inference pathway, such as analog-to-digital and digital-to-analog noise [Rasch et al., 2023, Lammie et al., 2024]. Finally, considerable work is needed to prove ff-EBM further at scale on more difficult tasks (e.g. standard ImageNet), considerably deeper architectures, and moving beyond vision tasks. One other exciting research direction would be the design of *ff-EBM based transformers*, with attention layers being chained with energy-based fully connected layers inside attention blocks.

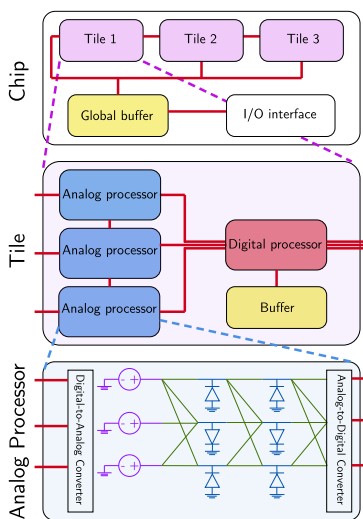

Figure 6: ff-EBMs as hierarchical systems implementing EP at chip scale (adapted from [Yi et al., 2023]) using *energy-based* analog processors made up of resistors (green edges), diodes (in blue), voltage sources (in purple), ADCs and DACs (adapted from [Scellier, 2024]), digital processors, memory buffers, all of these being connected by digital buses (red lines).

**Concluding remarks and broader impact.** We show that ff-EBMs constitute a novel framework for deep-learning in heterogeneous hardware settings. We hope that the proposed algorithm can help to overcome the typical division between digital *versus* analog or BP *versus* BP-free algorithms and that the greater energy-efficiency afforded by this framework provides a pragmatic, near-term blueprint to mitigating the dramatic carbon footprint of AI training [Strubell et al., 2020]. While we are still a long way from fully analog training accelerators at commercial maturity, we believe this work offers an incremental and sustainable roadmap to gradually integrate analog, energy-based computational primitives as they are developed into existing digital accelerators.

## Acknowledgements and disclosure of funding

The authors warmly thank Irina Rish, Jack Kendall and Suhas Kumar for their support of the project idea from the very start, Gregory Kollmer and Mohammed Fouda for helpful feedback on the manuscript as well as Benjamin Scellier for useful discussions last year which led to an alternative derivation of our main result (Appendix A.2.2). TN acknowledges the support from the Canada Excellence Research Chairs Program, as well as CIFAR and Union Neurosciences et Intelligence Artificielle Quebec (UNIQUE). This research was enabled by the computational resources provided by the Summit supercomputer, awarded through the Frontier DD allocation and INCITE 2023 program for the project "Scalable Foundation Models for Transferable Generalist AI" and SummitPlus allocation in 2024. These resources were supplied by the Oak Ridge Leadership Computing Facility at the Oak Ridge National Laboratory, with support from the Office of Science of the U.S. Department of Energy. ME acknowledges funding from Rain AI which commercializes technologies based on brain-inspired learning algorithms, as well as Constance Castres Saint-Martin for her unwavering support at the maternity hospital where most of this manuscript was written.

## Author contributions

TN was responsible for implementation, architecture design, coding all algorithmic details and running training experiments, as well as discovery of criteria for stable convergence. TN also participated in writing relevant portions of this manuscript. ME designed the study, derived all theoretical results, debugged and refactored the initial codebase and wrote most of the manuscript.

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

# A   Appendix

**Contents**

### A.1 Model details

#### A.1.1 Feedforward-tied EBMs (ff-EBMs)

We first formally define *Feedforward-tied Energy-based Models* (ff-EBMs) with precise assumptions on the energy-based and feedforward blocks.

**Definition A.1** (ff-EBMs). *A Feedforward-tied Energy-based Model (ff-EBM) of size $N$ comprises $N$ twice differentiable feedforward mapping $F^1, \cdots, F^N$ and $N-1$ twice differentiable energy functions $E^1, \cdots, E^{N-1}$ with respect to all their variables. For a given $x$, the inference procedure reads as:*

$$\begin{cases} s^0 := x \\ x_\star^k := F^k(s_\star^{k-1}, \omega^k), \quad s_\star^k := \arg\min_s E^k(s, \theta^k, x_\star^k) \quad \forall k = 1 \cdots N-1 \\ \hat{o}_\star := F^N(s_\star^{N-1}, \omega^N) \end{cases} \tag{17}$$

*Finally, we assume that $\forall k = 1 \cdots N-1$, $\nabla_1^2 E^k(s_\star^k, \theta^k, \omega^k)$ is invertible.*

#### A.1.2 Feedforward nets as a special case

We show that when energy-based blocks comprise a single layer only, the ff-EBM becomes purely feedforward.

**Lemma A.1.** *We consider ff-EBM per Def. (A.1) where the energy functions $E^k$ have the form:*

$$E^k(s^k, \theta^k, x^k) := G^k(s^k) - s^{k^\top} \cdot x^k + U^k(s^k, \theta^k). \tag{18}$$

*We assume that $U^k = 0$ for $k = 1 \cdots N-1$, $s \to \nabla G(s)$ is invertible and we denote $\sigma := \nabla G^{-1}$. Then, the resulting model is a feedforward model described by the following recursive equations:*

$$\begin{cases} s_\star^0 = x \\ x_\star^k = F^k(s_\star^{k-1}, \omega^k), \quad s_\star^k = \sigma(x_\star^k) \quad \forall k = 1 \cdots N-1 \\ \hat{o}_\star := F^N(s_\star^{N-1}, \omega^N) \end{cases} \tag{19}$$

*Proof of Lemma A.1.* Let $k \in [1, N-1]$. By definition of $s_\star^k$ and $x_\star^k$:

$$\nabla_1 E^k(s_\star^k, \theta^k, x_\star^k) = 0$$
$$\Leftrightarrow \quad \nabla G^k(s_\star^k) - x_\star^k + \nabla_1 U^k(s_\star^k, \theta^k) = 0$$
$$\Leftrightarrow \quad s_\star^k = \sigma\left(x_\star^k - \nabla_1 U^k(s_\star^k, \theta^k)\right) \tag{20}$$

Therefore Eq. (19) is immediately obtained from Eq. (20) with $U^k = 0$.

$\square$

#### A.1.3 Equilibrium computation

**For a single block.** As mentioned in Section 3.1, the energy function of the $k^{\text{th}}$ EB block has the form:

$$E^k(s^k, \theta^k, x^k) := G^k(s^k) - s^{k^\top} \cdot x^k + U^k(s^k, \theta^k), \tag{21}$$

where $x^k$ is the output of the preceding feedforward block. For a given choice of a continuously invertible activation function, $G_\sigma^k$ is defined as:

$$G_\sigma^k(s^k) := \sum_{i=1}^{\dim(s^k)} \int^{s_i} \sigma_i^{-1}(u_i) du_i \quad \text{such that} \quad \nabla G_\sigma^k(s^k)_i = \sigma_i^{-1}(s_i^k) \quad \forall i = 1 \cdots \dim(s^k). \tag{22}$$

To be more explicit and as we did previously, we re-write the augmented energy-function which encompasses both the $k^{\text{th}}$ EB block and the feedforward module that precedes it:

$$\widetilde{E}^k(s^k, \theta^k, s_\star^{k-1}, \omega^k) := E^k\left(s^k, \theta^k, F^k\left(s_\star^{k-1}, \omega^k\right)\right). \tag{23}$$

**Deep Hopfield Nets (DHNs) as EB blocks.**  In our experiments, we used weight matrices of the form:

$$\theta^k = \begin{bmatrix} 0 & \theta_1^{k^\top} & 0 & & \\ \theta_1^k & 0 & \theta_2^{k^\top} & & \\ 0 & \theta_2^k & \ddots & \ddots & \\ & & \ddots & 0 & \theta_L^{k^\top} \\ & & & \theta_L^k & 0 \end{bmatrix}, \tag{24}$$

whereby each layer $\ell$ is only connected to its adjacent neighbors. Therefore, fully connected and convolutional DHNs with $L$ layers have an energy function of the form:

$$U_{\text{FC}}^k(s^k, \theta^k) := -\frac{1}{2} s^{k^\top} \cdot \theta^k \cdot s^k = -\frac{1}{2} \sum_\ell s_{\ell+1}^{k^\top} \cdot \theta_\ell^k \cdot s_\ell^k \tag{25}$$

$$U_{\text{CONV}}^k(s^k, \theta^k) := -\frac{1}{2} s^k \bullet \left(\theta^k \star s^k\right) = -\frac{1}{2} \sum_\ell s_{\ell+1}^k \bullet \left(\theta_\ell^k \star s_\ell^k\right) \tag{26}$$

**Synchronous fixed-point iteration.**  We showed that when $G$ is chosen such that $\nabla G = \sigma^{-1}$ for some activation function $\sigma$, then the steady state of the $k^{\text{th}}$ block reads:

$$s_\star^k := \sigma\left(x^k - \nabla_1 U^k(s_\star^k, \theta^k)\right), \tag{27}$$

which justifies the following fixed-point iteration scheme, when the block is influenced by some error signal $\delta s$ with nudging strength $\beta$:

$$s_{\pm\beta,t+1}^k \leftarrow \sigma\left(x^k - \nabla_1 U^k(s_{\pm\beta,t}^k, \theta^k) \mp \beta \delta s^k\right). \tag{28}$$

The dynamics prescribed by Eq. 28 are also used for the inference phase with $\beta = 0$. To further refine Eq. (28), let us re-write Eq. (28) with a layer index $\ell$ where $\ell \in [1, L_k]$ with $L_k$ being the number of layers in the $k^{\text{th}}$ block, and replacing $x^k$ by its explicit expression:

$$\forall \ell = 1 \cdots L_k : \; s_{\ell,\pm\beta,t+1}^k \leftarrow \sigma\left(F^k\left(s_\star^{k-1}, \omega^{k-1}\right) - \nabla_{s_\ell^k} U^k(s_{\pm\beta,t}^k, \theta^k) \mp \beta \delta s^k\right). \tag{29}$$

As done in past EP works [Ernoult et al., 2019, Laborieux et al., 2021, Laborieux and Zenke, 2022, 2023, Scellier et al., 2024] and for notational convenience, we introduce the *primitive function* of the $k^{\text{th}}$ block as:

$$\Phi^k\left(s^k, \theta^k, s_\star^{k-1}, \omega^k\right) := s^{k^\top} \cdot F^k\left(s_\star^{k-1}, \omega^k\right) - U^k(s^k, \theta^k) \tag{30}$$

such that Eq. (29) re-writes:

$$\forall \ell = 1 \cdots L_k : s_{\ell,\pm\beta,t+1}^k \leftarrow \sigma\left(\nabla_{s_\ell^k} \Phi\left(s_{\pm\beta,t}^k, \theta^k, s_\star^{k-1}, \omega^k\right) \mp \beta \delta s^k\right). \tag{31}$$

Eq. (31) depicts a *synchronous* scheme where all layers are simultaneously updated at each timestep.

**Asynchronous fixed-point iteration.** Another possible scheme, employed by Scellier et al. [2024], instead prescribes to *asynchronously* update odd and even layers and was shown to speed up convergence in practice:

$$\begin{cases} \forall \text{ odd } \ell \in \{1, \cdots, L_k\}: & s_{\ell, \pm\beta, t+\frac{1}{2}}^k \leftarrow \sigma\left(\nabla_{s_\ell^k}\Phi\left(s_{\pm\beta, t}^k, \theta^k, s_\star^{k-1}, \omega^k\right) \mp \beta\delta s^k\right), \\ \forall \text{ even } \ell \in \{1, \cdots, L_k\}: & s_{\ell, \pm\beta, t+1}^k \leftarrow \sigma\left(\nabla_{s_\ell^k}\Phi\left(s_{\pm\beta, t+\frac{1}{2}}^k, \theta^k, s_\star^{k-1}, \omega^k\right) \mp \beta\delta s^k\right). \end{cases} \tag{32}$$

We formally depict this procedure as the subroutine `Asynchronous` inside Alg. 3. In practice, we observe that it was more practical to use a *fixed* number of iterations rather than using a convergence criterion with a fixed threshold.

---

**Algorithm 3** `Asynchronous` (for all blocks **until penultimate**)

*Input*: $T, \theta^k, \omega^k, s_\star^{k-1}, \beta, \delta s^k$
*Output*: $s_\beta^k$

1: $s^k \leftarrow 0$
2: **for** $t = 1 \cdots T$ **do**
3: $\quad \forall \text{ odd } \ell \in \{1, \cdots, L_k\}: s_{\ell,\beta}^k \leftarrow \sigma\left(\nabla_{s_\ell^k}\Phi\left(s_\beta^k, \theta^k, s_\star^{k-1}, \omega^k\right) - \beta\delta s^k\right)$
4: $\quad \forall \text{ even } \ell \in \{1, \cdots, L_k\}: s_{\ell,\beta}^k \leftarrow \sigma\left(\nabla_{s_\ell^k}\Phi\left(s_\beta^k, \theta^k, s_\star^{k-1}, \omega^k\right) - \beta\delta s^k\right)$
5: **end for**

---

**Resulting ff-EBM inference algorithm.** With the aforementioned details in hand, we re-write the inference algorithm Alg. 1 presented in the main as a `Forward` subroutine.

---

**Algorithm 4** `Forward`

*Input*: $T, x, W = \{\theta^1, \omega^1, \cdots \omega^N\}$
*Output*: $s^1, \cdots, s^{N-1}$ or $\hat{o}$ depending on the context

1: $s^0 \leftarrow x$
2: **for** $k = 1 \cdots N - 1$ **do**
3: $\quad s^k \leftarrow$ `Asynchronous` $\left(T, \theta^k, \omega^k, s^{k-1}\right)$ $\qquad\qquad\qquad\qquad\qquad\qquad$ ▷ Alg. 3
4: **end for**
5: $\hat{o} \leftarrow F^N\left(s, \omega^N\right)$

---

## A.2 Main theoretical result

### A.2.1 Proof of Theorem 3.1

The proof of Theorem 3.1 is structured as follows:

- We directly solve the multilevel problem optimization defined inside Eq. (9) using a Lagrangian-based approach (Lemma A.2), yielding optimal Lagrangian multipliers, block states and loss gradients.

- We show that by properly nudging the blocks, EP implicitly estimates the previously derived Lagrangian multipliers (Lemma A.3).

- We demonstrate Theorem 3.1 by combining Lemma A.2 and Lemma A.3.

- Finally, we highlight that when a ff-EBM is a feedforward net (Lemma A.1), then the proposed algorithm reduces to BP (Corollary A.1).

**Lemma A.2** (Lagrangian-based approach). *Assuming a ff-EBM (Def. A.1), we denote $s_\star^1, x_\star^1, \cdots, s_\star^{N-1}, \hat{o}_\star$ the states computed during the forward pass as prescribed by Eq. (17). Then, the gradients of the objective function $\mathcal{C} := \ell(\hat{o}(s_\star^{N-1}), y)$ as defined in the multilevel optimization problem (Eq. (9)), where it is assumed that $\ell$ is differentiable, read:*

$$
\begin{cases}
d_{\omega^N}\mathcal{C} = \partial_2 F^N(s_\star^{N-1}, \omega^N)^\top \cdot \partial_1 \ell(\hat{o}_\star, y), \\
d_{\theta^k}\mathcal{C} = \nabla^2_{1,2}\widetilde{E}^k(s_\star^k, \theta^k, s_\star^{k-1}, \omega^k) \cdot \lambda_\star^k \quad \forall k = 1 \cdots N-1, \\
d_{\omega^k}\mathcal{C} = \nabla^2_{1,4}\widetilde{E}^k(s_\star^k, \theta^k, s_\star^{k-1}, \omega^k) \cdot \lambda_\star^k \quad \forall k = 1 \cdots N-1,
\end{cases}
\tag{33}
$$

*where $\lambda_\star^1, \cdots, \lambda_\star^{N-1}$ satisfy the following conditions:*

$$
\begin{cases}
\nabla_{s^{N-1}}\ell(\hat{o}(s_\star^{N-1}), y) + \nabla^2_1\widetilde{E}^{N-1}(s_\star^{N-1}, \theta^{N-1}, s_\star^{N-2}, \omega^{N-1}) \cdot \lambda_\star^{N-1} = 0 \\
\forall k = N-2, \cdots, 1: \\
\quad \nabla^2_{1,3}\widetilde{E}^{k+1}\left(s_\star^{k+1}, \theta^{k+1}, s_\star^k, \omega^{k+1}\right) \cdot \lambda_\star^{k+1} + \nabla^2_1\widetilde{E}^k\left(s_\star^k, \theta^k, s_\star^{k-1}, \omega^k\right) \cdot \lambda_\star^k = 0
\end{cases}
\tag{34}
$$

*Proof of Lemma A.2.* Denoting $s := (s^1, \cdots, s^{N-1})^\top$ the state variables of the energy-based blocks, $\lambda := (\lambda^1, \cdots, \lambda^{N-1})^\top$ the Lagrangian multipliers associated with each of these variables, $W := \{\theta_1, \omega_1, \cdots, \theta_{N-1}, \omega_{N-1}\}$ the energy-based and feedforward parameters and $\hat{o}(s^{N-1}) := F^N\left(s^{N-1}, \omega^{N-1}\right)$ the logits, the Lagrangian of the multilevel optimization problem as defined in Eq. (9) reads:

$$
\mathcal{L}(s, \lambda, W) := \ell\left(\hat{o}(s^{N-1}), y\right) + \sum_{k=1}^{N-1} \lambda^{k\top} \cdot \nabla_1\widetilde{E}^k(s^k, \theta^k, s^{k-1}, \omega^k), \quad s^0 := x
\tag{35}
$$

Writing the associated Karush-Kuhn-Tucker (KKT) conditions $\partial_{1,2}\mathcal{L}(s_\star, \lambda_\star, W) := 0$ satisfied by optimal states and Lagrangian multipliers $s_\star$ and $\lambda_\star$, we get :

$$
\nabla_1\widetilde{E}^k(s_\star^k, \theta^k, s_\star^{k-1}, \omega^k) = 0 \quad \forall k = 1, \cdots, N-1
\tag{36}
$$

$$
\nabla_{s^{N-1}}\ell(\hat{o}(s_\star^{N-1}), y) + \nabla^2_1\widetilde{E}^{N-1}(s_\star^{N-1}, \theta^{N-1}, s_\star^{N-2}, \omega^{N-1}) \cdot \lambda_\star^{N-1} = 0
\tag{37}
$$

$$
\nabla^2_{1,3}\widetilde{E}^{k+1}\left(s_\star^{k+1}, \theta^{k+1}, s_\star^k, \omega^{k+1}\right) \cdot \lambda_\star^{k+1} + \nabla^2_1\widetilde{E}^k\left(s_\star^k, \theta^k, s_\star^{k-1}, \omega^k\right) \cdot \lambda_\star^k = 0 \quad \forall k = N-2, \cdots, 1
\tag{38}
$$

Eq. (36) governs the bottom-up block-wise relaxation procedure (as depicted in Alg. 1), while Eq. (37) and Eq. (38) governs error propagation in the last block and previous blocks respectively. Given $s_\star$ and $\lambda_\star$ by Eq. (36) – Eq. (38), the *total* derivative of the loss function with respect to the model parameters read:

$$d_W \ell(\hat{o}_\star, y) = d_W \left( \ell(\hat{o}_\star, y) + \sum_{k=1}^{N-1} \lambda_\star^{k\top} \cdot \underbrace{\nabla_1 \widetilde{E}^k(s_\star^k, \theta^k, s_\star^{k-1}, \omega^k)}_{=0 \quad \text{(Eq. (36))}} \right)$$

$$= d_W \mathcal{L}(s_\star, \lambda_\star, W)$$

$$= d_W s_\star^\top \cdot \underbrace{\partial_1 \mathcal{L}(s_\star, \lambda_\star, W)}_{=0 \quad \text{(Eq. (36))}} + d_W \lambda_\star^\top \cdot \underbrace{\partial_2 \mathcal{L}(s_\star, \lambda_\star, W)}_{=0 \quad \text{(Eq. (37)–(38))}} + \partial_3 \mathcal{L}(s_\star, \lambda_\star, W)$$

$$= \partial_3 \mathcal{L}(s_\star, \lambda_\star, W) \tag{39}$$

More precisely, applying Eq. (39) to the feedforward and energy-based block parameters yields:

$$d_{\omega^N} \ell(\hat{o}_\star, y) = \partial_2 F^N(s_\star^{N-1}, \omega^N)^\top \cdot \nabla_1 \ell(\hat{o}_\star, y),$$

$$d_{\theta^k} \ell(\hat{o}_\star, y) = \nabla_{1,2}^2 \widetilde{E}^k(s_\star^k, \theta^k, s_\star^{k-1}, \omega^k) \cdot \lambda_\star^k \quad \forall k = 1 \cdots N-1$$

$$d_{\omega^k} \ell(\hat{o}_\star, y) = \nabla_{1,4}^2 \widetilde{E}^k(s_\star^k, \theta^k, s_\star^{k-1}, \omega^k) \cdot \lambda_\star^k \quad \forall k = 1 \cdots N-1$$

$\square$

**Lemma A.3** (Computing Lagrangian multipliers by EP). *Under the same hypothesis as Lemma A.2, we define the nudged state of block $k$, denoted as $s_\beta^k$, implicitly through $\nabla_1 \mathcal{F}^k(s_\beta^k, \theta^k, x_\star^k, \delta s^k, \beta) = 0$ with:*

$$\mathcal{F}^k(s^k, \theta^k, x_\star^k, \delta s^k, \beta) := E^k(s^k, \theta^k, x_\star^k) + \beta s^{k\top} \cdot \delta s^k. \tag{40}$$

*Defining $(\delta s^k)_{k=1 \cdots N-1}$ recursively as:*

$$\delta s^{N-1} := \nabla_{s^{N-1}} \ell(\hat{o}_\star, y), \quad \delta s^k := d_\beta \left( \nabla_3 \widetilde{E}^{k+1}\left(s_\beta^{k+1}, \theta^{k+1}, s_\star^k, \omega^{k+1}\right) \right)\Big|_{\beta=0} \quad \forall k = 1 \cdots N-2, \tag{41}$$

*then we have:*

$$\lambda_\star^k = d_\beta \left(s_\beta^k\right)|_{\beta=0} \quad \forall k = 1 \cdots N-1, \tag{42}$$

*where $(\lambda_k)_{k=1 \cdots N-1}$ are the Lagrangian multipliers associated to the multilevel optimization problem defined in Eq. (9).*

*Proof of Lemma A.3.* We prove this result by backward induction on $k$.

**Initialization** ($k = N-1$). By definition, $s_\beta^{N-1}$ satisfies :

$$\beta \nabla_{s^{N-1}} \ell(\hat{o}_\star, y) + \nabla_1 \widetilde{E}^{N-1}\left(s_\beta^{N-1}, \theta^{N-1}, s_\star^{N-2}, \omega^{N-1}\right) = 0 \tag{43}$$

Differentiating Eq. (43) with respect to $\beta$ and evaluating the resulting expression at $\beta = 0$, we obtain:

$$\nabla_{s^{N-1}} \ell(\hat{o}_\star, y) + \nabla_1^2 \widetilde{E}^{N-1}\left(s_\star^{N-1}, \theta^{N-1}, s_\star^{N-2}, \omega^{N-1}\right) \cdot d_\beta s_\beta^{N-1}|_{\beta=0} = 0 \tag{44}$$

Substracting out Eq. (37) defining the Lagrangian multiplier $\lambda_\star^{N-1}$ and Eq. (44), we obtain:

$$\nabla_1^2 \widetilde{E}^{N-1}\left(s_\star^{N-1}, \theta^{N-1}, s_\star^{N-2}, \omega^{N-1}\right) \cdot \left(d_\beta s_\beta^{N-1}|_{\beta=0} - \lambda_\star^{N-1}\right) = 0 \tag{45}$$

By invertibility of $\nabla_1^2 \widetilde{E}^{N-1}\left(s_\star^{N-1}, \theta^{N-1}, s_\star^{N-2}, \omega^{N-1}\right)$, we therefore have that:

$$\lambda_\star^{N-1} = d_\beta s_\beta^{N-1}|_{\beta=0} \tag{46}$$

**Backward induction step** ($k + 1 \to k$). Let us assume that $\lambda_\star^{k+1} = d_\beta s_\beta^{k+1}|_{\beta=0}$. We want to prove that $\lambda_\star^k = d_\beta s_\beta^k|_{\beta=0}$. Again, $s_\beta^{k+1}$ satisfies by definition:

$$\beta \delta s^k + \nabla_1 \widetilde{E}^k \left( s_\beta^k, \theta^k, s_\star^{k-1}, \omega^k \right) = 0, \quad \delta s^k := d_\beta \left( \nabla_3 \widetilde{E}^{k+1} \left( s_\beta^{k+1}, \theta^{k+1}, s_\star^k, \omega^{k+1} \right) \right)\Big|_{\beta=0}. \tag{47}$$

On the one hand, proceeding as for the initialization step, differentiating Eq. (47) with respect to $\beta$ and taking $\beta = 0$ yields:

$$\delta s^k + \nabla_1^2 \widetilde{E}^k (s_\star^k, \theta^k, s_\star^{k-1}, \omega^k) \cdot d_\beta s_\beta^k|_{\beta=0} = 0. \tag{48}$$

On the other hand, note that $\delta s^k$ rewrites :

$$
\begin{aligned}
\delta s^k &= d_\beta \left( \nabla_3 \widetilde{E}^{k+1} \left( s_\beta^{k+1}, \theta^{k+1}, s_\star^k, \omega^{k+1} \right) \right)\Big|_{\beta=0} \\
&= \nabla_{1,3}^2 \widetilde{E}^{k+1} \left( s_\star^{k+1}, \theta^{k+1}, s_\star^k, \omega^{k+1} \right) \cdot ds_\beta^{k+1}\Big|_{\beta=0} \\
&= \nabla_{1,3}^2 \widetilde{E}^{k+1} \left( s_\star^{k+1}, \theta^{k+1}, s_\star^k, \omega^{k+1} \right) \cdot \lambda_\star^{k+1}, 
\end{aligned} \tag{49}
$$

where we used at the last step the recursion hypothesis. Therefore combining Eq. (48) and Eq. (49), we get:

$$\nabla_{1,3}^2 \widetilde{E}^{k+1} \left( s_\star^{k+1}, \theta^{k+1}, s_\star^k, \omega^{k+1} \right) \cdot \lambda_\star^{k+1} + \nabla_1^2 \widetilde{E}^k (s_\star^k, \theta^k, s_\star^{k-1}, \omega^k) \cdot d_\beta s_\beta^k|_{\beta=0} = 0. \tag{50}$$

Finally, we substract out Eq. (38) and Eq. (50) to obtain:

$$\nabla_1^2 \widetilde{E}^k (s_\star^k, \theta^k, s_\star^{k-1}, \omega^k) \cdot \left( d_\beta s_\beta^k|_{\beta=0} - \lambda_\star^k \right) = 0. \tag{51}$$

We conclude again by invertibility of $\nabla_1^2 \widetilde{E}^k (s_\star^k, \theta^k, s_\star^{k-1}, \omega^k)$ that $\lambda_\star^k = d_\beta s_\beta^k|_{\beta=0}$.

$\square$

**Theorem A.1** (Formal). *Assuming a model of the form Eq. (5), we denote $s_\star^1, x_\star^1, \cdots, s_\star^{N-1}, \hat{o}_\star$ the states computed during the forward pass as prescribed by Alg. 1. We define the nudged state of block $k$, denoted as $s_\beta^k$, implicitly through $\nabla_1 \mathcal{F}^k(s_\beta^k, \theta^k, x_\star^k, \delta s^k, \beta) = 0$ with:*

$$\mathcal{F}^k(s^k, \theta^k, x_\star^k, \delta s^k, \beta) := E^k(s^k, \theta^k, x_\star^k) + \beta {s^k}^\top \cdot \delta s^k. \tag{52}$$

*Denoting $\delta s^k$ and $\Delta x^k$ the error signals computed at the input of the feedforward block $F^k$ and of the energy-based block $E^k$ respectively, $g_{\theta^k}$ and $g_{\omega^k}$ the gradients of the loss function:*

$$\forall k = 1, \cdots, N-1 : \; g_{\theta^k} := d_{\theta^k} \mathcal{C}, \qquad \forall k = 1 \cdots N : \; g_{\omega^k} := d_{\omega^k} \mathcal{C}, \tag{53}$$

*then the following chain rule applies:*

$$\delta s^{N-1} := \nabla_{s^{N-1}} \ell(\hat{o}_\star, y), \quad g_{\omega^N} = \partial_2 F^N \left( s_\star^{N-1}, \omega^N \right)^\top \cdot \nabla_1 \ell(\hat{o}_\star, y) \tag{54}$$

$$\forall k = 1 \cdots N - 1 :$$

$$\begin{cases} \Delta x^k = d_\beta \left( \nabla_3 E^k(s_\beta^k, \theta^k, x_\star^k) \right)\Big|_{\beta=0}, & g_{\theta^k} = d_\beta \left( \nabla_2 E^k(s_\beta^k, \theta^k, x_\star^k) \right)\Big|_{\beta=0} \\ \delta s^{k-1} = \partial_1 F^k \left( s_\star^{k-1}, \omega^k \right)^\top \cdot \Delta x^k, & g_{\omega^k} = \partial_2 F^k \left( s_\star^{k-1}, \omega^k \right)^\top \cdot \Delta x^k \end{cases} \tag{55}$$

*Proof of Theorem A.1.* Combining Lemma A.2 and Lemma A.3, the following chain rule computes loss gradients correctly:

$$\delta s^{N-1} := \nabla_{s^{N-1}} \ell(\hat{o}_\star, y), \quad g_{\omega^N} = \partial_2 F^N \left(s_\star^{N-1}, \omega^N\right)^\top \cdot \nabla_1 \ell(\hat{o}_\star, y) \tag{56}$$

$$\forall k = 1 \cdots N - 1:$$

$$\begin{cases} \Delta s^{k-1} = d_\beta \left(\nabla_3 \widetilde{E}^k \left(s_\beta^k, \theta^k, s_\star^{k-1}, \omega^k\right)\right)\Big|_{\beta=0}, \quad g_{\theta^k} = \nabla_{1,2}^2 \widetilde{E}^k(s_\star^k, \theta^k, s_\star^{k-1}, \omega^k) \cdot d_\beta s_\beta^k|_{\beta=0} \\ g_{\omega^k} = \nabla_{1,4}^2 \widetilde{E}^k(s_\star^k, \theta^k, s^{k-1\star, \omega^k}) \cdot d_\beta s_\beta^k|_{\beta=0} \end{cases}$$

$$\tag{57}$$

Therefore to conclude the proof, we need to show that $\forall k = 1, \cdots, N - 1$:

$$d_\beta \left(\nabla_3 \widetilde{E}^k \left(s_\beta^k, \theta^k, s_\star^{k-1}, \omega^k\right)\right)\Big|_{\beta=0} = \partial_1 F^k \left(s_\star^{k-1}, \omega^k\right)^\top \cdot d_\beta \left(\nabla_3 E^k(s_\beta^k, \theta^k, x_\star^k)\right)\Big|_{\beta=0} \tag{58}$$

$$\nabla_{1,2}^2 \widetilde{E}^k(s_\star^k, \theta^k, s_\star^{k-1}, \omega^k) \cdot d_\beta s_\beta^k|_{\beta=0} = d_\beta \left(\nabla_2 E^k(s_\beta^k, \theta^k, x_\star^k)\right)\Big|_{\beta=0} \tag{59}$$

$$\nabla_{1,4}^2 \widetilde{E}^k(s_\star^k, \theta^k, s_\star^{k-1}, \omega^k) \cdot d_\beta s_\beta^k|_{\beta=0} = \partial_2 F^k \left(s_\star^{k-1}, \omega^k\right)^\top \cdot d_\beta \left(\nabla_3 E^k(s_\beta^k, \theta^k, x_\star^k)\right)\Big|_{\beta=0} \tag{60}$$

Let $k \in [1, N - 1]$. We prove Eq. (58) as:

$$\begin{aligned} d_\beta \left(\nabla_3 \widetilde{E}^k \left(s_\beta^k, \theta^k, s_\star^{k-1}, \omega^k\right)\right)\Big|_{\beta=0} &= d_\beta \left(\nabla_{s^{k-1}} E^k \left(s_\beta^k, \theta^k, F^k \left(s_\star^{k-1}, \omega^k\right)\right)\right)\Big|_{\beta=0} \\ &= \partial_1 F^k \left(s_\star^{k-1}, \omega^k\right)^\top \cdot d_\beta \left(\nabla_3 E^k(s_\beta^k, \theta^k, x_\star^k)\right)\Big|_{\beta=0} \end{aligned}$$

Eq. (59) can be obtained as:

$$\begin{aligned} \nabla_{1,2}^2 \widetilde{E}^k(s_\star^k, \theta^k, s_\star^{k-1}, \omega^k) \cdot d_\beta s_\beta^k|_{\beta=0} &= d_\beta \left(\nabla_2 \widetilde{E}^k(s_\beta^k, \theta^k, s_\star^{k-1}, \omega^k)\right)\Big|_{\beta=0} \\ &= d_\beta \left(\nabla_2 E^k(s_\beta^k, \theta^k, x_\star^k)\right)\Big|_{\beta=0} \end{aligned}$$

Finally and similarly, we have:

$$\begin{aligned} \nabla_{1,4}^2 \widetilde{E}^k(s_\star^k, \theta^k, s_\star^{k-1}, \omega^k) \cdot d_\beta s_\beta^k|_{\beta=0} &= d_\beta \left(\nabla_4 \widetilde{E}^k(s_\beta^k, \theta^k, s_\star^{k-1}, \omega^k)\right)\Big|_{\beta=0} \\ &= d_\beta \left(\nabla_{\omega^k} E^k(s_\beta^k, \theta^k, F^k \left(s_\star^{k-1}, \omega^k\right))\right)\Big|_{\beta=0} \\ &= d_\beta \left(\partial_2 F \left(s_\star^{k-1}, \omega^k\right)^\top \cdot \nabla_3 E^k(s_\beta^k, \theta^k, x_\star^k)\right)\Big|_{\beta=0} \\ &= \partial_2 F \left(s_\star^{k-1}, \omega^k\right)^\top \cdot d_\beta \left(\nabla_3 E^k(s_\beta^k, \theta^k, x_\star^k)\right)\Big|_{\beta=0} \end{aligned}$$

$$\square$$

### A.2.2 An alternative proof of Theorem 3.1

**An energy function for ff-EBMs?** While it is clear that the energy function of a ff-EBM is not $E = \sum_{k=1}^{N-1} \widetilde{E}^k$ (which would correspond in this case to the "single block" standard case), one may wonder if:

- ff-EBM inference (Alg. 1) can still be described as the minimization of some energy function?
- Therefore, if Theorem 3.1 can be derived by directly applying EP to this energy function?

We show below that this is indeed the case. We follow Zach [2021], denoting $s := (s^{1\top}, \cdots, s^{N-1\top})^\top$ and $W = \{W_1, \cdots, W_{N-1}\}$, by picking the following energy function:

$$
\begin{aligned}
\mathcal{F}(s, W, x, \beta) := \sum_{k=1}^{N-1} & \left\{ \widetilde{E}^k \left( s^k, W^k, s_\star^{k-1} \right) \right. \\
& \left. + \left[ \nabla_3 \widetilde{E}^{k+1} \left( s^{k+1}, W^{k+1}, s_\star^k \right) - \nabla_3 \widetilde{E}^{k+1} \left( s_\star^{k+1}, W^{k+1}, s_\star^k \right) \right]^\top \cdot \left( s^k - s_\star^k \right) \right\} \\
& + \widetilde{E}^{N-1} \left( s^{N-1}, W^{N-1}, s_\star^{N-2} \right) + \beta \widetilde{\ell}(s^{N-1}, y, W^N),
\end{aligned}
\tag{61}
$$

where we locally redefine $x$ as the concatenation of *all* block inputs, i.e. $x \leftarrow (x^\top, s_\star^{1\top}, \cdots, s_\star^{N-2\top})^\top$, and with $s_\star := (s_\star^{1\top}, \cdots, s_\star^{N-1\top})$ implicitly defined through $\nabla_1 \mathcal{F}(s_\star, W, x, \beta = 0) = 0$. In Lemma A.4, we show that the free steady state of the above energy function ($s_\star$ obtained with $\beta = 0$ inside Eq. (61)) indeed corresponds to the states computed by the ff-EBM inference scheme (Alg. 1).

---

**Lemma A.4.** *Let $\widetilde{E}^1, \cdots, \widetilde{E}^{N-1}$ be the block-wise energy functions of a ff-EBM defined per Def. A.1. Assume $s_\star$ implicitly defined through $\nabla_1 \mathcal{F}(s_\star, W, \beta = 0) = 0$ where $\mathcal{F}$ is defined by Eq. (61). Then:*

$$
s_\star^0 := x, \quad \forall k = 1, \cdots N - 1: \quad \nabla_1 \widetilde{E}^k(s_\star^k, W^k, s_\star^{k-1}) = 0
\tag{62}
$$

---

*Proof of Lemma A.4.* For $k = N - 1$, the stationarity condition $\nabla_{s^{N-1}} \mathcal{F}(s_\star, W, x, \beta)$ yields:

$$
\nabla_1 \widetilde{E}^{N-1} \left( s_\star^{N-1}, W^{N-1}, s_\star^{N-2} \right) + 0 = 0.
\tag{63}
$$

Then, for any $1 \le k < N - 1$, $\nabla_{s^k} \mathcal{F}(s_\star, W, x, \beta) = 0$ yields:

$$
\nabla_1 \widetilde{E}^k(s_\star^k, W^k, s_\star^{k-1}) + \underbrace{\left[ \nabla_3 \widetilde{E}^{k+1} \left( s_\star^{k+1}, W^{k+1}, s_\star^k \right) - \nabla_3 \widetilde{E}^{k+1} \left( s_\star^{k+1}, W^{k+1}, s_\star^k \right) \right]}_{=0} = 0
\tag{64}
$$

Eq. (63) and Eq. (63) indeed correspond to ff-EBM inference as depicted inside Alg. 1. $\qquad\square$

---

**The EP fundamental Lemma.** For self-containedness of this paper, we restate the fundamental EP result below inside Lemma A.5.

---

**Lemma A.5** ([Scellier, 2021]). *Let $\mathcal{F}(s, W, x, \beta)$ be a twice differentiable function of the three variables $s$, $W$ and $\beta$. For fixed $W$, $x$ and $\beta$, let $s_\beta$ be a point that satisfies the stationarity condition:*

$$
\nabla_1 \mathcal{F}(s_\beta, W, x, \beta) = 0,
\tag{65}
$$

*and suppose that $\nabla_1^2 \mathcal{F}(s_\beta, W, x, \beta)$ is invertible. Then, in the neighborhood of this point, we can define a continuously differentiable function $(x, W, \beta) \to s_\beta$ such that Eq. (65) holds for any $(x, W, \beta)$ in this neighborhood. Furthermore, we have the following identity:*

$$
d_W \left( \nabla_\beta \mathcal{F}(s_\beta, W, x, \beta) \right) = d_\beta \left( \nabla_2 \mathcal{F}(s_\beta, W, x, \beta) \right)
\tag{66}
$$

---

In particular, Eq. (66) may be evaluated with $\mathcal{F} = E + \beta \ell$ at $\beta = 0$ to yield the EP learning rule, denoting $\mathcal{C} := \ell(s_\star, y)$ [Scellier and Bengio, 2017]:

$$
d_W \mathcal{C} = d_\beta \left( \nabla_2 \mathcal{F}(s_\beta, W, x, \beta) \right)|_{\beta=0}
\tag{67}
$$

**Theorem 3.1 as a direct application of EP.** Now we know Eq. (61) defines a valid energy function for ff-EBMs and with Lemma A.5 in hand, we are ready to apply EP directly to this energy function. We rewrite below the block-wise free energy functions at use inside Theorem 3.1 and used in practice inside Alg. 2 to nudge a block of energy $\widetilde{E}^k$ given some top-down error signal $\delta^k$:

$$
\begin{cases}
\widetilde{\mathcal{F}}^k(s^k, W^k, s_\star^{k-1}, \delta s^k, \beta) := \widetilde{E}^k(s^k, W^k, s_\star^{k-1}) + \beta s^{k^\top} \cdot \delta s^k, \\
\delta s^k := d_\beta \left( \nabla_3 \widetilde{E}^{k+1} \left( s_\beta^k, W^{k+1}, s_\star^k \right) \right) \Big|_{\beta=0} \text{ if } k < N-1 \text{ else } \nabla_1 \widetilde{\ell}(s_\star^{N-1}, y, W^N)
\end{cases} \tag{68}
$$

In Theorem A.2, we show that the direct application of Lemma A.5 to $\mathcal{F}$ as defined inside Eq. (61) yields the same gradient formula for each parameter $W^k$ and the same nudged block states as those prescribed by Theorem 3.1 for sufficiently small $\beta$.

**Theorem A.2** (Informal). *Let $\mathcal{F}$ be defined as in Eq. (61) satisfying the same assumptions as in Lemma A.5. For fixed $W$, $x$ and $\beta$, let $s_\beta$ satisfy the stationarity condition:*

$$
\nabla_1 \mathcal{F}(s_\beta, W, x, \beta) = 0.
$$

*Then, we have:*

$$
d_{W^k}\mathcal{C} = d_\beta \left( \nabla_2 \widetilde{E}^k(s_\beta^k, W^k, s_\star^{k-1}) \right) \Big|_{\beta=0}, \quad \nabla_1 \widetilde{\mathcal{F}}^k(s_\beta^k, W^k, s_\star^{k-1}, \delta s^k, \beta) = \mathcal{O}(\beta^2) \tag{69}
$$

*Proof of Theorem A.2.* Lemma A.5 yielding:

$$
d_{W^k}\mathcal{C} = d_\beta \left( \nabla_{W^k} \mathcal{F}(s_\beta, W, x, \beta) \right)\big|_{\beta=0},
$$

proving Eq. (69) amounts to show that:

$$
d_\beta \left( \nabla_{W^k} \mathcal{F}(s_\beta, W, x, \beta) \right)\big|_{\beta=0} = d_\beta \left( \nabla_2 \widetilde{E}^k(s_\beta^k, W^k, s_\star^{k-1}) \right) \Big|_{\beta=0}, \tag{70}
$$

$$
\nabla_1 \widetilde{\mathcal{F}}^k(s_\beta^k, W^k, s_\star^{k-1}, \delta s^k, \beta) = \mathcal{O}(\beta^2) \tag{71}
$$

On the one hand, we have:

$$
\begin{aligned}
\nabla_{W^k} \mathcal{F}(s_\beta, W, x, \beta) = {} & \nabla_2 \widetilde{E}^k(s_\beta^k, W^k, s_\star^{k-1}) \\
& + \left( \nabla_{2,3}^2 \widetilde{E}^k(s_\beta^k, W^k, s_\star^{k-1}) - \nabla_{2,3}^2 \widetilde{E}^k(s_\star^k, W^k, s_\star^{k-1}) \right) \cdot \left( s_\beta^{k-1} - s_\star^{k-1} \right)
\end{aligned} \tag{72}
$$

For notational convenience, we define $A(\beta) := \left( \nabla_{2,3}^2 \widetilde{E}^k(s_\beta^k, W^k, s_\star^{k-1}) - \nabla_{2,3}^2 \widetilde{E}^k(s_\star^k, W^k, s_\star^{k-1}) \right)$. Note that $A(\beta = 0) = 0$. Differentiating Eq. (72) with respect to $\beta$ and taking $\beta = 0$ yields:

$$
\begin{aligned}
d_\beta \left( \nabla_{W^k} \mathcal{F}(s_\beta, W, x, \beta) \right)|_{\beta=0} = {} & d_\beta \left( \nabla_2 \widetilde{E}^k(s_\beta^k, W^k, s_\star^{k-1}) \right)|_{\beta=0} \\
& + d_\beta A(\beta)|_{\beta=0} \cdot \underbrace{\left( s_{\beta=0}^{k-1} - s_\star^{k-1} \right)}_{=0} + \underbrace{A(0)}_{=0} \cdot \left( s_\beta^{k-1} - s_\star^{k-1} \right) \\
= {} & d_\beta \left( \nabla_2 \widetilde{E}^k(s_\beta^k, W^k, s_\star^{k-1}) \right)|_{\beta=0},
\end{aligned}
$$

which proves Eq. (70).

On the other hand, the stationary condition $\nabla_{s^k} \mathcal{F}(s_\beta, W, x, \beta)$ on the last block ($k = N-1$) yields:

$$
\begin{aligned}
& \nabla_1 \widetilde{E}^{N-1}(s_\beta^{N-1}, W^{N-1}, s_\star^{N-2}) + \beta \nabla_1 \widetilde{\ell}(s_\beta^{N-1}, y, W^N) = 0 \\
\Rightarrow {} & \nabla_1 \widetilde{E}^{N-1}(s_\beta^{N-1}, W^{N-1}, s_\star^{N-2}) + \beta \nabla_1 \widetilde{\ell}(s_\star^{N-1}, y, W^N) = \mathcal{O}(\beta^2) \\
\Leftrightarrow {} & \nabla_1 \widetilde{\mathcal{F}}^{N-1}(s_\beta^{N-1}, W^{N-1}, s_\star^{N-2}, \delta s^N, \beta) = \mathcal{O}(\beta^2).
\end{aligned} \tag{73}
$$

For previous blocks, i.e. $k < N - 1$, we have:

$$\nabla_{s^k} \mathcal{F}(s_\beta, W, x, \beta) = 0$$

$$\Leftrightarrow \nabla_1 \tilde{E}^k(s_\beta^k, W^k, s_\star^{k-1}) + \nabla_3 \widetilde{E}^{k+1}\left(s_\beta^{k+1}, W^{k+1}, s_\star^k\right) - \nabla_3 \widetilde{E}^{k+1}\left(s_\star^{k+1}, W^{k+1}, s_\star^k\right) = 0$$

$$\Rightarrow \nabla_1 \tilde{E}^k(s_\beta^k, W^k, s_\star^{k-1}) + d_\beta \left(\nabla_3 \widetilde{E}^{k+1}\left(s_\beta^{k+1}, W^{k+1}, s_\star^k\right)\right)\Big|_{\beta=0} = \mathcal{O}(\beta^2)$$

$$\Leftrightarrow \nabla_1 \widetilde{\mathcal{F}}^k(s_\beta^k, W^k, s_\star^{k-1}, \delta s^k, \beta) = \mathcal{O}(\beta^2). \tag{74}$$

Altogether, Eq. (73) and Eq. (74) finishes to prove Eq. (71). $\qquad\square$

### A.3 Resulting algorithms

#### A.3.1 Explicit BP-EP chaining

We presented in Alg. 2 a "pure" EP algorithm where the BP-EP gradient chaining is *implicit*. We show below, inside Alg. 5, an alternative implementation (equivalent in the limit $\beta \to 0$) where the use of BP through feedforward modules is *explicit* and which is the direct implementation of Theorem A.1. We also show the resulting algorithm when the ff-EBM reduces to a feedforward net (Lemma A.1) inside Alg. 7, highlight in blue the statements which differ from the general case presented inside Alg. 5.

---

**Algorithm 5** Explicit BP-EP gradient chaining (Theorem (3.1))

---

1: $\delta s, g_{\omega^N} \leftarrow \nabla_{s^{N-1}}\ell(\hat{o}_\star, y), \nabla_{\omega^N}\ell(\hat{o}_\star, y)$ ▷ Single backprop step
2: **for** $k = N - 1 \cdots 1$ **do**
3: $\quad s_\beta \leftarrow \underset{s}{\textbf{Optim}} \left[ E^k(s, \theta^k, x_\star^k) + \beta s^\top \cdot \delta s \right]$ ▷ EP through $E^k$
4: $\quad s_{-\beta} \leftarrow \underset{s}{\textbf{Optim}} \left[ E^k(s, \theta^k, x_\star^k) - \beta s^\top \cdot \delta s \right]$
5: $\quad g_{\theta^k} \leftarrow \frac{1}{2\beta} \left( \nabla_2 E^k(s_\beta, \theta^k, x_\star^k) - \nabla_2 E^k(s_{-\beta}, \theta^k, x_\star^k) \right)$
6: $\quad \Delta x \leftarrow \frac{1}{2\beta} \left( \nabla_3 E^k(s_\beta, \theta^k, x_\star^k) - \nabla_3 E^k(s_{-\beta}, \theta^k, x_\star^k) \right)$
7: $\quad g_{\omega^k} \leftarrow \partial_2 F^k\left(s_\star^{k-1}, \omega^k\right)^\top \cdot \Delta x$ ▷ Explicit BP through $F^k$
8: $\quad \delta s \leftarrow \partial_1 F^k\left(s_\star^{k-1}, \omega^k\right)^\top \cdot \Delta x$
9: **end for**

---

#### A.3.2 Recovering backprop through feedforward nets as a special case

**Corollary A.1.** *Under the same hypothesis as Theorem A.1 and Lemma A.1, then the following chain rule applies to compute error signals backward from the output layer:*

$$\begin{cases} \delta s^{N-1} := \nabla_{s^{N-1}}\ell(\hat{o}_\star, y), \quad g_{\omega^N} = \nabla_{\omega^N}\ell(\hat{o}_\star, y) \\ \Delta x^k = \sigma'(x_\star^k) \odot \delta s^k \\ \delta s^{k-1} = \partial_1 F^k\left(s_\star^{k-1}, \omega^k\right)^\top \cdot \Delta x^k, \quad g_{\omega^k} = \partial_2 F^k\left(s_\star^{k-1}, \omega^k\right)^\top \cdot \Delta x^k \end{cases} \tag{75}$$

*Proof of Corollary A.1.* Let $k \in [1, N-1]$. As we can directly apply Theorem A.1 here, proving the result simply boils down to showing that:

$$\Delta x^k = \sigma'(x_\star^k) \odot \delta s^k \tag{76}$$

First, we notice that when $E^k$ is of the form of Eq. (18), then $\Delta x^k$ reads as:

$$\Delta x^k = d_\beta\left(\nabla_3 E^k(s_\beta^k, \theta^k, x_\star^k)\right)\big|_{\beta=0} = - d_\beta\left(s_\beta^k\right)\big|_{\beta=0}. \tag{77}$$

$s_\beta^k$ satisfies, by definition and when $U^k = 0$:

$$\sigma^{-1}(s_\beta^k) - x_\star^k + \beta\delta s^k = 0$$
$$\Leftrightarrow \quad s_\beta^k = \sigma\left(x_\star^k - \beta\delta s^k\right) \tag{78}$$

Combining Eq. (77) and Eq. (78) yields Eq. (76), and therefore, along with Theorem A.1, the chain-rule Eq. (75). $\qquad\square$

We showcase in Alg. 6 and Alg. 7 the resulting algorithms implicit and explicit BP-EP chaining respectively, with lines in blue highlighting differences with the general algorithm Alg. 2.

**Algorithm 6** Implicit BP-EP gradient chaining with $U^k = 0$

---

1: $\delta s, g_{\omega^N} \leftarrow \nabla_{s^{N-1}} \ell(\hat{o}_\star, y), \nabla_{\omega^N} \ell(\hat{o}_\star, y)$ ▷ Single backprop step
2: **for** $k = N-1 \cdots 1$ **do**
3:      $s_\beta,\ s_{-\beta} \leftarrow \sigma\left(x_\star^k - \beta \delta s^k\right), \sigma\left(x_\star^k + \beta \delta s^k\right)$ ▷ EP through $\widetilde{E}^k$
4:      $g_{\omega^k} \leftarrow \frac{1}{2\beta}\left(\nabla_4 \widetilde{E}^k(s_\beta, \theta^k, s_\star^{k-1}, \omega^k) - \nabla_4 \widetilde{E}^k(s_{-\beta}, \theta^k, s_\star^{k-1}, \omega^k)\right)$ ▷ i-BP through $F^k$
5:      $\delta s \leftarrow \frac{1}{2\beta}\left(\nabla_3 \widetilde{E}^k(s_\beta, \theta^k, s_\star^{k-1}, \omega^k) - \nabla_3 \widetilde{E}^k(s_{-\beta}, \theta^k, s_\star^{k-1}, \omega^k)\right)$
6: **end for**

---

**Algorithm 7** Explicit BP-EP gradient chaining with $U^k = 0$

---

1: $\delta s, g_{\omega^N} \leftarrow \nabla_{s^{N-1}} \ell(\hat{o}_\star, y), \nabla_{\omega^N} \ell(\hat{o}_\star, y)$ ▷ Single backprop step
2: **for** $k = N-1 \cdots 1$ **do**
3:      $\Delta x \leftarrow -\frac{1}{2\beta}\left(\sigma\left(x_\star^k - \beta \delta s^k\right) - \sigma\left(x_\star^k + \beta \delta s^k\right)\right)$
4:      $g_{\omega^k} \leftarrow \partial_2 F^k\left(s_\star^{k-1}, \omega^k\right)^\top \cdot \Delta x$ ▷ Explicit BP through $F^k$
5:      $\delta s \leftarrow \partial_1 F^k\left(s_\star^{k-1}, \omega^k\right)^\top \cdot \Delta x$
6: **end for**

---

### A.3.3   Detailed implementation of the implicit BP-EP chaining algorithm (Alg. 2)

**Nudging the last block.** From looking at the procedure prescribed by Theorem 3.1 and algorithms thereof (Alg. 2, Alg. 5), all the error signals used to nudge the EB blocks are *stationary*, including the top-most block where the loss error signal is fed in. Namely, the augmented energy function of the last block reads as:

$$\mathcal{F}^{N-1}(s^{N-1}, \theta^{N-1}, x_\star^{N-1}, \beta) := E^{N-1}(s^{N-1}, \theta^{N-1}, x_\star^{N-1}) + \beta {s^{N-1}}^\top \cdot \nabla_{s^{N-1}} \ell(\hat{o}_\star, y), \quad (79)$$

where $\hat{o}_\star := F^N\left(s_\star^{N-1}, \omega^N\right)$ is *constant*. Up to a constant, Eq. (80) uses the cost function *linearized around* $s_\star^{N-1}$ instead of the cost function itself. This is, however, in contrast with most EP implementations where the nudging force acting upon the EB block is usually *elastic*, i.e. the nudging depends on the current state of the EB block. More precisely, instead of using Eq. (79), we instead use:

$$\mathcal{F}^{N-1}(s^{N-1}, \theta^{N-1}, x_\star^{N-1}, \beta) := E^{N-1}(s^{N-1}, \theta^{N-1}, x_\star^{N-1}) + \beta \ell(F^N(s^{N-1}, \omega^N), y), \quad (80)$$

This results in the following asynchronous fixed-point dynamics for the last block:

$$\begin{cases} \forall \text{ odd } \ell \in \{1, \cdots, L_k\}: & s_{\ell, \pm\beta, t+\frac{1}{2}}^k \leftarrow \sigma\left(\nabla_{s_\ell^k} \Phi\left(s_{\pm\beta, t}^k, \theta^k, s_\star^{k-1}, \omega^k\right) \mp \beta \nabla_{s^k} \ell(s_{\pm\beta, t}^k, y)\right), \\ \forall \text{ even } \ell \in \{1, \cdots, L_k\}: & s_{\ell, \pm\beta, t+1}^k \leftarrow \sigma\left(\nabla_{s_\ell^k} \Phi\left(s_{\pm\beta, t+\frac{1}{2}}^k, \theta^k, s_\star^{k-1}, \omega^k\right) \mp \beta \nabla_{s^k} \ell(s_{\pm\beta, t}^k, y)\right). \end{cases}$$

The resulting `Asynchronous` subroutine, applying for the last block, is depicted inside Alg. 8.

**Readout.** Laborieux et al. [2021] introduced the idea of the "readout" whereby the last linear layer computing the loss logits is *not* part of the EB free block dynamics but simply "reads out" the state of the penultimate block. In all our experiments we use such a readout in combination with the cross entropy loss function. Using our formalism, our readout is simply the last feedforward transformation used inside $\ell$, namely $F^N(\cdot, \omega^N)$.

**Detailed implicit EP-BP chaining algorithm.** We provide a detailed implementation of our algorithm presented in the main (Alg. 2) in Alg. 11. As usually done for EP experiments, we always perform a "free phase" to initialize the block states (`Forward` subroutine, Alg. 4). Then, two

---

**Algorithm 8** `Asynchronous` (for **last** block)

---

*Input*: $T, \theta^{N-1}, \omega^{N-1}, \omega^N, s_\star^{k-1}, \beta, \ell$ (cost function), $y$
*Output*: $s_\beta^{N-1}$

1: $s^{N-1} \leftarrow 0$
2: **for** $t = 1 \cdots T$ **do**
3:      $\forall$ odd $\ell \in \{1, \cdots, L_N\}$:
4:         $s_{\ell,\beta}^{N-1} \leftarrow \sigma \left( \nabla_{s_\ell^{N-1}} \Phi \left( s_\beta^{N-1}, \theta^{N-1}, s_\star^{N-2}, \omega^{N-1} \right) - \beta \nabla_{s_\ell^{N-1}} \ell(F^N(s^{N-1}, \omega^N), y) \right)$
5:      $\forall$ even $\ell \in \{1, \cdots, L_N\}$:
6:         $s_{\ell,\beta}^{N-1} \leftarrow \sigma \left( \nabla_{s_\ell^{N-1}} \Phi \left( s_\beta^{N-1}, \theta^{N-1}, s_\star^{N-2}, \omega^{N-1} \right) - \beta \nabla_{s_\ell^{N-1}} \ell(F^N(s^{N-1}, \omega^N), y) \right)$
7: **end for**

---

nudged phases are applied to the last block and parameter gradients subsequently computed, as done classically (`BlockGradient` subroutine for the last block, Alg. 9), with an extra computation to compute the error current to be applied to the penultimate block ($\delta s^{N-2}$). Then, the same procedure is recursed backward through blocks (Alg. 10), until reaching first block.

---

**Algorithm 9** `BlockGradient` (for **last** block)

---

*Input*: $T, s_\star^{N-2}, \theta^{N-1}, \omega^{N-1}, \omega^N, \beta, \ell, y$
*Output*: $\delta s^{N-2}$

1: $s_\beta^{N-1} \leftarrow$ `Asynchronous` $\left( T, \theta^{N-1}, \omega^{N-1}, \omega^N, \beta, \ell, y \right)$          ▷ Alg. 8
2: $s_{-\beta}^{N-1} \leftarrow$ `Asynchronous` $\left( T, \theta^{N-1}, \omega^{N-1}, \omega^N, -\beta, \ell, y \right)$
3: $g_{\omega^N} \leftarrow \frac{1}{2} \left( \nabla_{s^{N-1}} \ell(F^N \left( s_\beta^{N-1}, \omega^N \right)) + \nabla_{s^{N-1}} \ell(F^N \left( s_{-\beta}^{N-1}, \omega^N \right)) \right)$
4: $g_{\theta^{N-1}} \leftarrow \frac{1}{2\beta} \left( \nabla_2 \widetilde{E}^{N-1}(s_\beta^{N-1}, \theta^{N-1}, s_\star^{N-2}, \omega^{N-1}) - \nabla_2 \widetilde{E}^{N-1}(s_{-\beta}^{N-1}, \theta^{N-1}, s_\star^{N-2}, \omega^{N-2}) \right)$
5: $g_{\omega^{N-1}} \leftarrow \frac{1}{2\beta} \left( \nabla_4 \widetilde{E}^{N-1}(s_\beta^{N-1}, \theta^{N-1}, s_\star^{N-2}, \omega^{N-1}) - \nabla_4 \widetilde{E}^{N-1}(s_{-\beta}^{N-1}, \theta^{N-1}, s_\star^{N-2}, \omega^{N-2}) \right)$
6: $\delta s^{N-2} \leftarrow \frac{1}{2\beta} \left( \nabla_3 \widetilde{E}^{N-1}(s_\beta^{N-1}, \theta^{N-1}, s_\star^{N-2}, \omega^{N-1}) - \nabla_3 \widetilde{E}^{N-1}(s_{-\beta}^{N-1}, \theta^{N-1}, s_\star^{N-2}, \omega^{N-2}) \right)$

---

**Algorithm 10** `BlockGradient` (for all blocks **until penultimate**)

---

*Input*: $T, s_\star^{k-1}, \theta^k, \omega^k, \beta, \delta s$
*Output*: $\delta s^{k-1}$

1: $s_\beta^k \leftarrow$ `Asynchronous` $\left( T, \theta^k, \omega^k, \beta, \delta s \right)$          ▷ Alg. 3
2: $s_{-\beta}^k \leftarrow$ `Asynchronous` $\left( T, \theta^k, \omega^k, -\beta, \delta s \right)$
3: $g_{\theta^k} \leftarrow \frac{1}{2\beta} \left( \nabla_2 \widetilde{E}^k(s_\beta^k, \theta^k, s_\star^{k-1}, \omega^k) - \nabla_2 \widetilde{E}^k(s_{-\beta}^k, \theta^k, s_\star^{k-1}, \omega^k) \right)$
4: $g_{\omega^k} \leftarrow \frac{1}{2\beta} \left( \nabla_4 \widetilde{E}^k(s_\beta^k, \theta^k, s_\star^{k-1}, \omega^k) - \nabla_4 \widetilde{E}^k(s_{-\beta}^k, \theta^k, s_\star^{k-1}, \omega^k) \right)$
5: $\delta s^{k-1} \leftarrow \frac{1}{2\beta} \left( \nabla_3 \widetilde{E}^k(s_\beta^k, \theta^k, s_\star^{k-1}, \omega^k) - \nabla_3 \widetilde{E}^k(s_{-\beta}^k, \theta^k, s_\star^{k-1}, \omega^k) \right)$

---

**Algorithm 11** Detailed implicit BP-EP gradient chaining

---

1: $s_\star^1, \cdots, s_\star^{N-1} \leftarrow$ `Forward` $\left( T_{\text{free}}, x, W \right)$          ▷ Alg. 4
2: $\delta s \leftarrow$ `BlockGradient` $\left( T_{\text{nudge}}, s_\star^{N-2}, \theta^{N-1}, \omega^{N-1}, \omega^N, \beta, \ell, y \right)$          ▷ Alg. 9
3: **for** $k = N-2 \cdots 1$ **do**
4:      $\delta s \leftarrow$ `BlockGradient` $\left( T_{\text{nudge}}, s_\star^{k-1}, \theta^k, \omega^k, \beta, \delta s \right)$          ▷ Alg. 10
5: **end for**

---

## A.4 Static gradient analysis

**Important foreword.** The whole subsection is dedicated to an important tool when developing code for EP research. While EP is agnostic to *how* the steady states are obtained – the EP theory only prescribes they are energy minimizers – they can be obtained in practice (i.e. *in simulations*) through fixed-point iteration schemes (see Appendix A.1.1). The below formally defines the computational graph spanned by these schemes and abstract them away into a transition function $K$ and defines three different techniques to compute gradients on this graph: *Automatic Differentiation* (AD, Prop. A.1), *Implicit Differentiation* (ID, Def. A.3) or *Equilibrium Propagation* (EP, Def. A.4). After defining each of these algorithms formally, we will state and demonstrate an equivalence between EP and ID (Theorem A.3) which we test numerically and relied upon for the development of our codebase.

### A.4.1 Algorithmic baselines

**Definition of the computational graph being optimized.** We abstract fixed-point iteration dynamics away into a *kernel function* $K$ which, given some block state $s_t^k$ yields $s_{t+1}^k$.

**Definition A.2** (Form of the computational graph through equilibrium)**.**

$$\forall k = 1, \cdots, N-1, \ \forall t = 1, \cdots, \tau :$$
$$x^0 = x, \quad s_t^k = K(s_{t-1}^k, W_{t-1}^k = W^k, x^k = s_\tau^{k-1}), \quad \mathcal{C} = \ell(F^N(s_\tau^{N-1}, \omega^N), y) := \tilde{\ell}(s^{N-1}, y) \tag{81}$$

Note that we emphasize, through the $W_{t-1}^k = W^k$ notation, that the parameters $W^k$ are shared across *all timesteps* $t = 1, \cdots, \tau$. This will help us define loss gradient with respect to $W_{t-1}^k$ further below, *i.e.* how much $W^k$ contributes *at time* $t-1$ to changing the loss $\mathcal{C}$. The *total* contribution of $W^k$ reads as the sum of the elemental contributions of all $W_t^k$. This intuition is more precisely illustrated further below. Given the computational graph defined in Def. A.2, we can now formally define the *Automatic Differentiation* (AD) baseline.

**Automatic Differentiation (AD).** Our goal is to compute:

$$g_{W^k}^{\mathrm{AD}} := \hat{g}_{W^k}^{\mathrm{AD}}(\tau) \quad \text{with:} \quad \hat{g}_{W^k}^{\mathrm{AD}}(t) := \sum_{k=1}^{t} \partial_{W_{\tau-k}^k} \mathcal{C} \tag{82}$$

In plain words, $\hat{g}_{W^k}^{\mathrm{AD}}(t)$ denotes the loss gradient for parameter $W^k$ *truncated at the $t^{\mathrm{th}}$ step* moving backward in time. We formally define below *Automatic Differentiation* (AD).

**Proposition A.1** (Automatic Differentiation (AD))**.** *The gradients $\hat{g}_{W^k}^{\mathrm{AD}}(t)$ can be computed using the following recursive equations:*

$$\forall k = N-1 \cdots 1 :$$
$$\delta s_0^k = \delta x_\tau^{k+1} \ \text{if } k < N-1 \ \text{else } \nabla_1 \tilde{\ell}(s_\tau^{N-1}, y)$$
$$\delta x_0^k = 0, \quad \hat{g}_{W^k}^{\mathrm{AD}}(0) = 0$$
$$\forall t = 1, \cdots, \tau :$$
$$\begin{cases} \delta s_t^k = \partial_1 K(s_{\tau-t}^k, W^k, x^k = s_\tau^{k-1})^\top \cdot \delta s_{t-1}^k \\ \hat{g}_{W^k}^{\mathrm{AD}}(t) = \hat{g}_{W^k}^{\mathrm{AD}}(t-1) + \partial_2 K(s_{\tau-t}^k, W^k, x^k = s_\tau^{k-1})^\top \cdot \delta s_{t-1}^k \\ \delta x_t^k = \delta x_{t-1}^k + \partial_3 K(s_{\tau-t}^k, W^k, x^k = s_\tau^{k-1})^\top \cdot \delta s_{t-1}^k \end{cases} \tag{83}$$

*Proof of Prop. A.1.* This is a straightforward application of the chain rule applied to Eq. (81). $\qquad \square$

**Implicit Differentiation (ID).** We define the steady state of block $k$, which we denote $s_\star^k$, as the fixed point of Eq. (81). With this notation in hand, we can define *Implicit Differentiation* (ID) in this setting.

**Definition A.3** (Implicit Differentiation (ID)). *Denoting $s^k_\star$ the fixed point of Eq. (81) inside block $k$, we define Implicit Differentiation (ID) through the following recursive equations:*

$$\forall k = N - 1 \cdots 1 :$$
$$\delta s^k_0 = \delta x^{k+1}_\tau \text{ if } k < N - 1 \text{ else } \nabla_1 \tilde{\ell}(s^{N-1}_\tau, y)$$
$$\delta x^k_0 = 0, \quad \hat{g}^{\text{ID}}_{W^k}(0) = 0$$
$$\forall t = 1, \cdots, \tau :$$
$$\begin{cases} \delta s^k_t = \partial_1 K(s^k_\star, W^k, x^k = s^{k-1}_\star)^\top \cdot \delta s^{k-1}_{t-1} \\ \hat{g}^{\text{ID}}_{W^k}(t) = \hat{g}^{\text{ID}}_{W^k}(t-1) + \partial_2 K(s^k_\star, W^k, x^k = s^{k-1}_\star)^\top \cdot \delta s^k_{t-1} \\ \delta x^k_t = \delta x^k_{t-1} + \partial_3 K(s^k_\star, W^k, x^k = s^{k-1}_\star)^\top \cdot \delta s^k_{t-1} \end{cases} \quad (84)$$

We are now ready to state a simple algorithmic equivalence between ID and AD, which we built upon for our implementation of Alg. 12.

**Corollary A.2** (Equivalence of ID and AD). *Assuming that:*

$$\forall k = 1, \cdots, N - 1, \ \forall t = 1, \cdots, \tau : \quad s^k_t = s^k_\star, \quad (85)$$

*where $s^k_\star$ denotes the fixed-point of Eq. 81, then automatic differentiation (Prop. A.1) and implicit differentiation (Def. A.3) are equivalent, namely:*

$$\forall k = 1, \cdots, N - 1, \ \forall t = 1, \cdots, \tau : \quad \hat{g}^{\text{ID}}_{W^k}(t) = \hat{g}^{\text{AD}}_{W^k}(t) \quad (86)$$

*Proof of Corollary A.2.* This is a straightforward application of the definition of AD (Prop. A.1 along with the hypothesis made inside Corollary A.2. □

**Resulting implementation of ID.** We describe our implementation of ID inside Alg. 12. First, we relax all blocks sequentially to equilibrium following Alg. 4 and we do not track gradients throughout this first phase, using $T_{\text{free}}$ fixed-point iteration steps per block. *Then*, initializing the block states with those computed at the previous step, we re-execute the same procedure (still with Alg. 4), this time *tracking gradients* and using $T_{\text{nudge}}$ steps fixed-point iteration steps for each block. Then, we use automatic differentiation to backpropagate through the last $T_{\text{nudge}}$ steps for each block, namely backpropagating, backward in time, *through equilibrium*.

---

**Algorithm 12** Our implementation of ID

---

1: Without tracking gradients:                                               ▷ e.g. with torch.no_grad()
2:   $s^1_\star, \cdots, s^{N-1}_\star \leftarrow \text{Forward}(T_{\text{free}}, x, W)$                              ▷ Alg. 4
3: Initialize block states at $s^1_\star, \cdots, s^{N-1}_\star$
4: $\hat{o}_\star \leftarrow \text{Forward}(T_{\text{nudge}}, x, W)$                               ▷ This time gradients are tracked
5: $\mathcal{C} \leftarrow \ell(\hat{o}_\star, y)$
6: Backpropagate $\mathcal{C}$ backward through the last $T_{\text{nudge}}$ steps for each block   ▷ e.g. C.backward()

---

**An important note about this implementation of ID.** Note that this is *not* a standard implementation of ID and it may be surprising at first glance to implement ID as AD, thereby loosing the constant $\mathcal{O}(1)$ memory cost of ID with respect to the length of the computational graph. Instead, the memory cost of Alg. 12 is $\mathcal{O}((N-1)\tau)$ [4]. However, our goal is not so much to optimize for memory usage (as in the context of Deep Equilibrium Models [Bai et al., 2019]) but to code an algorithmic baseline which we know to be equivalent to EP. Lastly, note that this implementation of ID is also known as *Recurrent Backprop* (RBP, [Almeida, 1987, Pineda, 1987]) or *Von-Neumann* RBP [Liao et al., 2018], and that ID generally comes in many more algorithmic flavors [Blondel et al., 2022].

---

[4] We are not accounting for the *spatial depth* ($L$) of the computational graph in this cost. In this case, standard ID would have memory cost $\mathcal{O}(L)$ and our implementation inside Alg. 12 $\mathcal{O}(L(N-1)\tau)$.

### A.4.2 Proof of Theorem 4.1

In order to state a formal equivalence between EP and ID, we first need to formally define EP in the context of the aforementioned computational graph defined in Def. A.2.

**Definition A.4** (Equilibrium Propagation (EP)). *Denoting $s_\star^k$ the fixed point of Eq. (81) inside block $k$ and assuming that the transition kernel $K$ has the form $K(s, W^k, x^k) = \nabla_1\Phi(s, W^k, x^k)$, we define Equilibrium Propagation (EP) through the following recursive equations:*

$$\forall k = N - 1 \cdots 1:$$
$$\delta s^k = \Delta x_\tau^{k+1} \text{ if } k < N - 1 \text{ else } \nabla_1\tilde{\ell}(s_\tau^{N-1}, y)$$
$$\Delta x_0^k = 0, \quad \hat{g}_{W^k}^{\text{EP}}(0) = 0, \quad s_{\beta,t=0}^k = s_\star^k$$
$$\forall t = 1, \cdots, \tau:$$
$$\begin{cases} s_{\beta,t+1}^k = \nabla_1\Phi(s_{\beta,t}^k, W^k, x^k = s_\star^{k-1}) - \beta\delta s^k \\ \hat{g}_{W^k}^{\text{EP}}(\beta, t) = -\frac{1}{2\beta}\left(\nabla_2\Phi(s_{\beta,t+1}^k, W^k, x^k = s_\star^{k-1}) - \nabla_2\Phi(s_{-\beta,t+1}^k, W^k, x^k = s_\star^{k-1})\right) \\ \Delta x_{\beta,t}^k = -\frac{1}{2\beta}\left(\nabla_3\Phi(s_{\beta,t+1}^k, W^k, x^k = s_\star^{k-1}) - \nabla_3\Phi(s_{-\beta,t+1}^k, W^k, x^k = s_\star^{k-1})\right) \end{cases}$$

Now that we have properly defined ID and EP, we are ready to state the main result of this section about the algorithmic equivalence between ID and EP which our coding work significantly built upon.

**Theorem A.3** (Extension of [Ernoult et al., 2019] to ff-EBMs). *Assuming that:*

$$\forall k = 1, \cdots, N - 1, \forall t = 1, \cdots, \tau: \quad s_t^k = s_\star^k, \tag{87}$$

*where $s_\star^k$ denotes the fixed-point of Eq. (81) and that the transition kernel $K$ has the form $K(s, W^k, x^k) = \nabla_1\Phi(s, W^k, x^k)$, then implicit differentiation (Def. A.3) and equilibrium propagation (Def. A.4) are equivalent in the limit $\beta \to 0$, namely:*

$$\forall k = 1, \cdots, N - 1, \forall t = 1, \cdots, \tau: \quad \lim_{\beta \to 0} \hat{g}_{W^k}^{\text{EP}}(\beta, t) = \hat{g}_{W^k}^{\text{ID}}(t) \tag{88}$$

*Proof of Theorem A.3.* This proof follows the exact same methodology as that of Ernoult et al. [2019]. For self-containedness though and because of some subtle differences, we carry out here the derivation. We first define:

$$\Delta s_t^k := d_\beta s_{t+1}^k|_{\beta=0} - d_\beta s_t^k|_{\beta=0}. \tag{89}$$

Note that since $s_{\beta,t=0}^k = s_\star$, $d_\beta s_t^k|_{\beta=0} = 0$ since $s_\star$ does not depend on $\theta$. Furthermore, note that by differentiating the equation satisfied by $s_{\beta,t+1}^k$ with respect to $\beta$ and evaluating the resulting expressions at $\beta = 0$ yields:

$$d_\beta s_{\beta,t+1}^k|_{\beta=0} = \partial_1 K(s_\star^k, W^k, x^k = s_\star^{k-1}) \cdot d_\beta s_{\beta,t}^k|_{\beta=0} - \delta s^k \tag{90}$$

In particular, evaluating Eq. (90) for $t = 0$ yields:

$$\Delta s_0^k = d_\beta s_{\beta,1}^k|_{\beta=0} - \underbrace{d_\beta s_{\beta,0}^k|_{\beta=0}}_{=0} = -\delta s^k. \tag{91}$$

Therefore, substracting Eq. (90) across two timesteps yields altogether:

$$\begin{aligned} \Delta s_t^k &= \partial_1 K(s_\star^k, W^k, x^k = s_\star^{k-1}) \cdot \Delta s_{t-1}^k \\ &= \nabla_1^2\Phi(s_\star^k, W^k, x^k = s_\star^{k-1}) \cdot \Delta s_{t-1}^k \\ &= \nabla_1^2\Phi(s_\star^k, W^k, x^k = s_\star^{k-1})^\top \cdot \Delta s_{t-1}^k \\ &= \partial_1 K(s_\star^k, W^k, x^k = s_\star^{k-1})^\top \cdot \Delta s_{t-1}^k \end{aligned} \tag{92}$$

Note that $\hat{g}^{\mathrm{EP}}_{W^k}(t)$ rewrites:

$$
\begin{aligned}
\hat{g}^{\mathrm{EP}}_{W^k}(\beta, t) &= -d_\beta\left(\nabla_2 \Phi(s^k_{\beta,t+1}, W^k, x^k = s^{k-1}_\star)\right) + \mathcal{O}(\beta^2)\\
&= -\nabla^2_{1,2}\Phi(s^k_\star, W^k, x^k = s^{k-1}_\star) \cdot d_\beta s^k_{\beta,t+1}|_{\beta=0} + \mathcal{O}(\beta^2)\\
&= -\nabla^2_{1,2}\Phi(s^k_\star, W^k, x^k = s^{k-1}_\star) \cdot \Delta s^k_t - \nabla^2_{1,2}\Phi(s^k_\star, W^k, x^k = s^{k-1}_\star) \cdot d_\beta s^k_{\beta,t}|_{\beta=0} + \mathcal{O}(\beta^2)\\
&= -\underbrace{\nabla^2_{1,2}\Phi(s^k_\star, W^k, x^k = s^{k-1}_\star)}_{=\nabla^2_{2,1}\Phi(s^k_\star, W^k, x^k = s^{k-1}_\star)^\top} \cdot \Delta s^k_t + \hat{g}^{\mathrm{EP}}_{W^k}(\beta, t-1) + \mathcal{O}(\beta^2)\\
&= \partial_2 K(s^k_\star, W^k, x^k = s^{k-1}_\star)^\top \cdot \left(-\Delta s^k_t\right) + \hat{g}^{\mathrm{EP}}_{W^k}(\beta, t-1) + \mathcal{O}(\beta^2)
\end{aligned}
\tag{93}
$$

Likewise, we can show that:

$$
\Delta x^k_{\beta,t} = \partial_3 K(s^k_\star, W^k, x^k = s^{k-1}_\star)^\top \cdot \left(-\Delta s^k_t\right) + \Delta x^k_{\beta,t-1} + \mathcal{O}(\beta^2)
\tag{94}
$$

Altogether, Eq. (91), Eq. (92) Eq. (93) and Eq. (94) yield, denoting $\hat{g}^{\mathrm{EP}}_{W^k}(t) := \lim_{\beta\to 0}\hat{g}^{\mathrm{EP}}_{W^k}(\beta, t)$ and $\Delta x^k_t := \lim_{\beta\to 0}\Delta x^k_{\beta,t}$:

$$
\begin{aligned}
&\forall k = N-1, \cdots, 1 :\\
&-\Delta s^k_0 = \delta s^k = \Delta x^{k+1}_\tau \text{ if } k < N-1 \text{ else } \nabla_1 \tilde{\ell}(s^{N-1}_\tau, y),\ \hat{g}^{\mathrm{EP}}_{W^k}(0) = 0,\ \Delta x^k_0 = 0\\
&\forall t = 1, \cdots, \tau :
\end{aligned}
\tag{95}
$$

$$
\begin{cases}
-\Delta s^k_t &= \partial_1 K(s^k_\star, W^k, x^k = s^{k-1}_\star)^\top \cdot (-\Delta s^k_{t-1})\\
\hat{g}^{\mathrm{EP}}_{W^k}(t) &= \hat{g}^{\mathrm{EP}}_{W^k}(t-1) + \partial_2 K(s^k_\star, W^k, x^k = s^{k-1}_\star)^\top \cdot \left(-\Delta s^k_t\right)\\
\Delta x^k_t &= \partial_3 K(s^k_\star, W^k, x^k = s^{k-1}_\star)^\top \cdot \left(-\Delta s^k_t\right) + \Delta x^k_{t-1}
\end{cases}
\tag{96}
$$

Starting from $k = N-1$, $(-\Delta s^{N-1}_t)_{t\in[\![1,\tau]\!]}$ and $(\delta s^{N-1}_t)_{t\in[\![1,\tau]\!]}$, $(\Delta x^{N-1}_t)_{t\in[\![1,\tau]\!]}$ and $(\delta x^{N-1})_{t\in[\![1,\tau]\!]}$, $(\hat{g}^{\mathrm{EP}}_{W^{N-1}}(t))_{t\in[\![1,\tau]\!]}$ and $(\hat{g}^{\mathrm{ID}}_{W^{N-1}}(t))_{t\in[\![1,\tau]\!]}$ satisfy the same initial conditions and recursive equations, therefore there are all (pair-wise) equal for $t = 1, \cdots, \tau$. Therefore in particular, $\Delta x^{N-1}_\tau = \delta x^{N-1}$ such that $(-\Delta s^{N-2}_\tau)_{t\in[\![1,\tau]\!]}$ and $(\delta s^{N-2}_t)_{t\in[\![1,\tau]\!]}$ from the previous $(N-2)^{\mathrm{th}}$ block satisfy the same initial conditions, such that the reasonning applying to $k = N-1$ recurses for $k < N-1$, which yields Eq. (88).

$\square$

### A.4.3 Details about Fig. 3

**Precise hyperparameters to reproduce Fig. 5 and Fig. 3 can be found inside our repository**.
Fig. 7 precisely depict the architecture at use for these experiments.

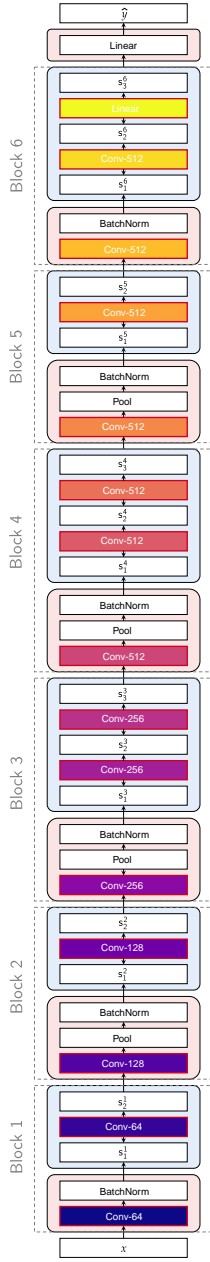

Figure 7: Architecture used for the static gradient analysis. The color code used to label layers matches that of Fig. 3 and Fig. 5. In the context of the static gradient analysis, "block" $k$ is defined as all layers participating in $\tilde{E}^k$, which therefore includes $F^k$ *and* $E^k$ modules (rather than one of these taken alone).

## A.5 Experimental Details

### A.5.1 Datasets

Simulations were run on CIFAR-10, CIFAR-100 and Imagenet32 datasets, all consisting of color images of size $32 \times 32$ pixels. CIFAR-10 [Krizhevsky, 2009] includes 60,000 color images of objects and animals. Images are split into 10 classes, with 6,000 images per class. Training data and test data include 50,000 images, and 10,000 images respectively. CIFAR-100 [Krizhevsky, 2009] likewise comprises 60,000 and features a diverse set of objects and animals split into 100 distinct classes. Each class contains 600 images. Like CIFAR-10, the dataset is divided into a training set with 50,000 images and a test set containing the remaining 10,000 images. The ImageNet32 dataset [Chrabaszcz et al., 2017] is a downsampled version of the original ImageNet dataset Russakovsky et al. [2015] containing 1,000 classes with 1,281,167 training images, 50,000 validation images, 100,000 test images and 1000 classes.

### A.5.2 Data preprocessing

All data were normalized according to statistics shown in 3 and augmented with 50% random horizontal flips. Images were randomly cropped and padded with the last value along the edge of the image.

Table 3: Data Normalization. Input images were normalized by conventional mean ($\mu$) and standard deviation ($\sigma$) values for each dataset. All images used are color (three channels).

| Dataset | Mean ($\mu$) | Standard deviation ($\sigma$) |
|---|---|---|
| CIFAR-10/100 | (0.4914, 0.4822, 0.4465) | (0.2470, 0.2435, 0.2616) |
| Imagenet32 | (0.485, 0.456, 0.406) | (0.3435, 0.336, 0.3375) |

### A.5.3 Simulation details

**Weight initialization.** EP, similar to other machine learning paradigms reliant on fixed-point iteration [Bai et al., 2019], is highly sensitive to initialization statistics [Agarwala and Schoenholz, 2022], hence conventionally difficult to tune, and requiring many iterations for the three relaxation phases. Initialization of weights as *Gaussian Orthogonal Ensembles* (GOE) ensures better stability (reduced variance) and, combined with other stabilizing measures discussed below, empirically yields faster convergence.
According to GOE, weights are intialized as:

$$W_{ij} \sim \begin{cases} \mathcal{N}(0, \frac{V}{N}), & \text{if } i \neq j \\ \mathcal{N}(0, \frac{2V}{N}), & \text{if } i = j \end{cases}$$

where $\mathcal{N}(\mu, \sigma^2)$ denotes a Gaussian (normal) distribution with mean $\mu$ and variance $\sigma^2$. $N$ was manually tuned for each architecture.

**State initialization.** All layers are initialized as zero matrices.

**Activation functions.** An important detail for faithful reproduction of these experiments is the choice and placement of activation functions applied during the iterative fixed-point procedure. In the literature, activations (i.e. "clamping") is conventionally applied at each layer, *with the exception of the final layer*, where it is sometimes included e.g. Scellier et al. [2024], and sometimes omitted Laborieux et al. [2021], depending on the loss function at use. For these experiments we used both the standard hard activation employed by Ernoult et al. [2019] and Scellier et al. [2024], and the more conservative one given in [Laborieux et al., 2021]. For the tolerance based and splitting experiments, we generalize the approach of Laborieux et al. [2021], by scaling values by a variable factor $\alpha$ instead of a fixed value $0.5$ . Details are given in Table 4.

In practice, we find that the smaller scaling factors corresponding with the "laborieux" activation, in conjunction with GOE, and the omission of clamping at the output of *each block* significantly

Table 4: Activation functions

| Name | Description | Source |
|------|-------------|--------|
| ernoult | $\sigma(x) = \max(\min(x, 1), 0)$ | [Ernoult et al., 2019] |
| laborieux | $\sigma(x) = \max(\min(0.5 \times x, 1), 0)$ | [Laborieux et al., 2021] |
| nest | $\sigma(x) = \max(\min(\alpha \times x, 1), 0)$ | This work |

enhances gradient stability and speeds convergence in deep multi-block settings. In the interest of multi-scale uniformity and consistency with previous literature [Laborieux et al., 2021] Ernoult et al. [2019], we apply clamping activations on *all layers* in our 6-layer architecture.

For the scaling experiments, we apply the "laborieux" activation at every layer *except* the output of each block. For the 12-layer splitting experiment, we do the same, omitting clamping from the output of the final layer of *each* block in the block-size=4 and block-size=3 experiments. However, in the block-size=2 case we clamp the output of the second and fourth blocks to preserve dynamics of the block-size=4 split. Such consistency is not possible for the block-size=3 experiment, constituting a possible discrepancy in dynamics.

**Cross-entropy loss and softmax readout.**   Following [Laborieux et al., 2021], all models were implemented such that the output $y$ is removed from the system (e.g. not included in the relaxation dynamics) but is instead the function of a weight matrix: $W_{\text{out}}$ of size $\dim(y) \times \dim(s)$, where $s$ is the state of the final layer. For each time-step $t$, $\hat{y}_t = \text{softmax}(W_{\text{out}} s_t)$.

The cross-entropy cost function associated with the softmax readout is then:

$$l(s, y, W_{\text{out}}) = -\sum_{c=1}^{C} y_c \log(\text{softmax}_c(W_{\text{out}} \cdot s)).$$

**Convention to count layers.**   It is important to note that by convention we refer to architectures throughout this text to the exclusion of the softmax readout, which is technically an additional layer, though not involved in the relaxation process.

**Architecture.**   All convolutional layers used in experiments are of kernel size 3 and stride and padding 1. Max-pooling was applied with a window of $2 \times 2$ and stride of 2. For the 6-layer model used in Table 1 , batchnorm was applied *after* the first layer convolution and pooling operation. All other models in both experiments use batch-normalization on the first layer of each block *after* convolution and pooling (where applied). These details exclude the linear softmax readout of size $n$ classes.

**Hyperparameters.**   **Detailed hyperparameters for to reproduce Table 1 and Table 2 are given inside the configuration files of our repository**. Note that all architectural details for the 12-layer models are *identical* across splitting and scaling experiments. Additionally, identical hyperparameters were used for CIFAR100 and Imagenet experiments of Table 2. Unlike previous literature, the use of GOE intialization eliminates the need for separate layerwise learning rates and initialization parameters. One noteworthy detail is that only 100 epochs were used for the larger model for Table 2 compared with 200 epochs for the smaller 12-layer model. This was due to prohibitively long run-time of training the larger model. Notably, performance still significantly improves with decreased overall runtime.

**Root-finding algorithms.**   While in principle any root-finding algorithm may be used for the two relaxation phases of our EP implementation (for inference and gradient computation), our implementation utilized a simple fixed-point iteration procedure, in which neuron states are initialized as zero vectors with values updated asynchronously on each iteration to that of the gradient of the total system energy with respect to current state. An approximate illustration of this procedure is found in Alg. 3. As indicated in Section 4.3, two variants of the convergence procedure were employed, one in which the average value of current state is compared to that of the previous state for each layer, and relaxation is truncated when values for all layers have a difference of *less than* 1e-4. This was known

as the tolerance-based (TOL) procedure. Notably, tolerance-based convergence criteria were applied *on the free phase only*, with nudging computed *with a fixed value of iterations*. This was to ensure consistency between ID and EP, though in practice a tolerance can be applied equally to the nudging phase.

---

**Algorithm 13** `Asynchronous with Tolerance` (for all blocks **until penultimate**)

---

*Input*: $T, \theta^k, \omega^k, s_\star^{k-1}, \beta, \delta s^k$
*Output*: $s_\beta^k$

1: $s^k \leftarrow 0$
2: $c \leftarrow \infty$
3: **for** $t = 1 \cdots T$ **do**
4:      $\forall$ odd $\ell \in \{1, \cdots, L_k\}:\ s_{\ell,\beta,temp}^k \leftarrow \sigma\left(\nabla_{s_\ell^k}\Phi\left(s_\beta^k, \theta^k, s_\star^{k-1}, \omega^k\right) - \beta\delta s^k\right)$
5:      $\forall$ even $\ell \in \{1, \cdots, L_k\}:\ s_{\ell,\beta,temp}^k \leftarrow \sigma\left(\nabla_{s_\ell^k}\Phi\left(s_\beta^k, \theta^k, s_\star^k, \omega^k\right) - \beta\delta s^k\right)$
6:      **if** $t \geq 2$ **then**
7:          $\forall \ell \in \{1, \cdots, L_k\}: c_\ell^k \leftarrow \frac{s_{\ell,\beta,temp}^k - s_{\ell,\beta}^k}{|s_{\ell,\beta}^k|}$
8:          **if** mean$(c^k) \leq Tol$ **then**
9:             **BREAK;**
10:          **end if**
11:      **end if**
12:      $s_{\ell,\beta}^k \leftarrow s_{\ell,\beta,temp}^k$
13: **end for**

---

**Supplementary results with a fixed number of iterations.** In addition to the TOL-based procedure, we obtained results for 4.3 using the more conventional approach of [Scellier and Bengio, 2019][Laborieux et al., 2021][Ernoult et al., 2019], applying fixed number of iterations on the first and second relaxation phases (see 1). This approach was also the default used for our scaling experiments in 4.4. Importantly, with the TOL procedure described above Alg 3 becomes Alg 13. Results using a fixed iteration root-finding scheme are shown in 5

Table 5: Validation accuracy and Wall Clock Time (WCT) obtained on CIFAR-10 by EP (Alg. 2) and ID on models with different number of layers ($L$) and block sizes ("bs"). 3 seeds are used.

| | EP | | ID | |
|---|---|---|---|---|
| | Top-1 (%) | WCT | Top-1 (%) | WCT |
| **L =6** | | | | |
| bs=6 | $88.8^{\pm 0.2}$ | 8:06 | $87.3^{\pm 0.6}$ | 8:05 |
| bs=3 | $89.5^{\pm 0.2}$ | 8:01 | $89.2^{\pm 0.2}$ | 7:40 |
| bs=2 | $90.1^{\pm 0.2}$ | 7:47 | $90.0^{\pm 0.2}$ | 7:18 |
| **L =12** | | | | |
| bs=4 | $91.6^{\pm 0.1}$ | 7:49 | $91.6^{\pm 0.1}$ | 7:08 |
| bs=3 | $92.2^{\pm 0.2}$ | 6:06 | $92.2^{\pm 0.1}$ | 5:59 |
| bs=2 | $91.7^{\pm 0.2}$ | 6:10 | $91.8^{\pm 0.1}$ | 6:08 |

**Other details.** All experiments were run using Adam optimizer [Kingma and Ba, 2014]and Cosine Annealing scheduler[Loshchilov and Hutter, 2017], specifying some minimum learning rates and setting maximum T equal to epochs (i.e. no warm restarts). Code was implemented in Pytorch 2.0 and all simulations were run on NVIDIA A100 SXM4 40GB GPUs.

