# OpenReview forum: "Towards training digitally-tied analog blocks via hybrid gradient computation"
_NeurIPS.cc/2024/Conference — NeurIPS 2024 spotlight_

### Official Review · Reviewer_pevw · 2024-06-21

**Soundness:** 3
**Presentation:** 3
**Contribution:** 3
**Rating:** 6
**Confidence:** 3

**Summary:**

State-of-the-art (SOTA) analog hardware accelerators consist of both analog and digital components supporting major and auxiliary operations. Moreover, they typically suffer from device imperfections on their analog parts. In this paper, the authors propose feedforward-tied energy-based models (ff-EBMs) for digital and analog circuits. ff-EBMs compute gradients by both backpropagating on feedforward parts and eq-propagating on energy-based parts. In this paper, ff-EBMs use a Deep Hopfield Network (DHN) as one example, which can be arbitrarily partitioned into uniform size. ff-EBMs achieves a SOTA performance on ImageNet32.

**Strengths:**

1. The paper works on an important and interesting problem.
2. The paper flows well.

**Weaknesses:**

1. The paper does NOT consider a detailed analog computing device model. It will be interesting to see how significant variations on the analog devices the ff-EBMs can tolerate. And what is the relationship between the convergence speed and the device variations?

2. The paper does not include energy-related data. What is the energy saving for the energy-based models? What is the energy saving of skipping the accurate gradient computations?

**Questions:**

Please comment on the points in the weakness section.

**Limitations:**

no limitations.

---

> ### Author Rebuttal · Authors · 2024-08-06
>
> We thank Reviewer pevw for their detailed review and we are happy that they've appreciated our work.
>
> i) *The paper does NOT consider a detailed analog computing device model*.
>
> We fully agree with Reviewer pevw that this is a current limitation of our work, as explicitly acknowledged inside the “Limitations and future work” paragraph (L.294-301). This is why moving towards **hardware realistic ff-EBM training is one of the main thrusts of our research trajectory**  - see section 3 of the proposed roadmap to Reviewer kDhg as well as Fig. 3 inside the pdf attached with the global rebuttal).
>
> This being rightfully pointed out, there are still good reasons not to start the investigation of ff-EBM training by EP with realistic analog computing device model but instead Deep Hopfield Networks (DHNs). DHNs, while being remotely inspired by physics [1] and without “analog-realistic” noise, serve as a useful model baseline to start with in the context of EP for at least three reasons:
>
> - In all past studies investigating the scalability of EP training and motivated by analog computing, DHNs were used as EB models [2, 3, 4, 5].
>
> - DHNs are *already difficult to train by EP*. Compared to feedforward models trained by backprop, training DHNs by EP comes with extra critical and subtle hyperparameter tuning, most importantly: a) the root find algorithm at use to compute equilibria [6, 7, 8]; b) the number of iterations needed to converge in the first, second and third phases of EP; c) the value of the nudging parameter $\beta$; d) the way nudging is applied in the output layer [9]; e) weight initialization statistics, which implicit learning more generally is highly sensitive to [6, 7]. For instance, our study reveals that initializing the weights of the EB blocks as *Gaussian Orthogonal Ensembles* (abbreviated as “GOE weight initialization” in our paper) was instrumental in having ff-EBM training by EP working properly.
>
> - Finally, simulating energy-based analog resistive networks comes with even more hyperparameters compared to DHNs [8, 10], making in turn EP training harder to debug.
>
> ii)	*How much can ff-EBMs tolerate significant variations on the analog devices?*
>
> We would like to clarify the scope of the question by defining two broad classes of analog device imperfections which Reviewer pevw may have in mind:
>
> - *Device imperfections affecting the **weight update** given a gradient value*. In analog systems, weights are typically encoded as conductance values of resistive devices [10]. *Given a gradient estimate*, programming a resistive device to update its conductance by this gradient estimate *precisely* is generally very hard, if not impossible because conductance updates suffer from device-to-device (“static”) **variability**, cycle-to-cycle (“dynamic”) **variability**, conductance drift, non-linearity with respect to the applied programming voltage, asymmetry between device potentiation and depression, etc. [11]. However, these imperfections **apply to any gradient computation algorithm** since they only pertain to the conductance update *given a gradient estimate*. As highlighted in the most recent study of “analog SGD” [14], the gradient estimate is *given*. Therefore, all conclusions regarding “analog parameter optimization” *given a gradient value* hold in particular for ff-EBM training by EP.
>
> - *Device imperfections affecting the **gradient estimation***. Once an analog nonlinear resistive circuit is instantiated with some *fixed* conductance values, it may still be subject to *static variability* (i.e. two analog devices do not share the exact same input-to-output characteristic) and *dynamic variability* (i.e. a single analog device may not react exactly the same when exposed twice to the same input). However, as highlighted in the global rebuttal and that addressed to Reviewer Jt5z, **EP-based training of EB blocks is inherently tolerant to this static variability** since this very same circuit is used for *both* inference *and* gradient computation [12]. Therefore, **ff-EBM training by EP may only be impacted by dynamic variability**. Investigating quantitatively the tolerance of ff-EBM training to dynamic variability and coming up with strategies to mitigate its potential negative effects is a project of its own that naturally falls in our research agenda (see task 3c) inside rebuttal to Reviewer kDhg and in Fig. 3 inside the pdf attached to the global rebuttal.
>
> iii)	*What is the relationship between the convergence speed and the device variations?*
> We assume that  Reviewer pevw has in mind the convergence of *the learning dynamics of ff-EBMs **in the parameter space*** when trained by the proposed algorithm. The “device variations” may refer in this case to those involved in the *conductance updates* which, as mentioned above, are *agnostic to how gradients are computed*. It is known that the learning dynamics of "analog SGD" [14] are severely impacted by such non idealities even on simple tasks as MNIST [13]. Therefore, any such observation, or theoretical prediction [14] as well as any heuristic to mitigate non idealities of the conductance updates, e.g. Tiki-Taka [15], *are also directly applicable to ff-EBM training by EP*.
>
> iv)	*What is the energy saving for the energy-based models? What is the energy saving of skipping the accurate gradient computations?*
> Please, see our global rebuttal.
>
>
> [1] *Hopfield, Science, 1986*
> [2] *Laborieux et al, Frontiers in Neuroscience, 2021*
> [3] *Scellier et al, 2022*
> [4] *Laborieux & Zenke, NeurIPS 2022*
> [5] *Laborieux & Zenke, ICLR 2024*
> [6] *Bai et al, NeurIPS 2019*
> [7] *Agarwala and Schoenholz, ICML 2022*
> [8] *Scellier, ICML 2024*
> [9] *Scellier et al, NeurIPS 2023*
> [10] *Kendall et al, 2020*
> [11] *Burr et al, Advance in Physics X, 2017*
> [12] *Yi et al, Nature Electronics, 2022*
> [13] *Burr et al, IEEE Transactions on Electron Devices, 2015*
> [14] *Wu et al, ArXiV, 2024*
> [15] *Gokmen & Haensch, Frontiers in Neurosciences, 2020*

---

### Official Review · Reviewer_kDhg · 2024-07-04

**Soundness:** 3
**Presentation:** 3
**Contribution:** 3
**Rating:** 7
**Confidence:** 4

**Summary:**

This paper discusses a novel approach to improving the power efficiency of AI training by integrating analog and digital hardware. The authors introduce a hybrid model called Feedforward-tied Energy-based Models (ff-EBMs) that combines feedforward neural networks with energy-based models (EBMs). This model aims to leverage the benefits of both analog and digital systems to create a more efficient computational framework for neural network training.

The paper introduces a new model, ff-EBMs, which integrates feedforward and EB modules. These models are designed to perform end-to-end gradient computation, combining backpropagation through feedforward blocks and "eq-propagation" through EB blocks. And bilevel and multi-level optimization are concepts used to frame the learning process of the Feedforward-tied Energy-based Models (ff-EBMs).

The paper promises to demonstrate the effectiveness of the proposed approach experimentally, using DHNs as EB blocks and achieving new state-of-the-art performance on the ImageNet32 dataset.

**Strengths:**

- The paper presents a robust framework that leverages the strengths of bilevel and multi-level optimization to address the challenges of training neural networks in a hybrid digital-analog setting. One of the significant strengths lies in the rigorous mathematical proofs provided for the bilevel optimization inherent to energy-based models (EBMs). The authors extend this foundation to a multi-level optimization approach, which is not only theoretically sound but also practically applicable to the training of Feedforward-tied Energy-based Models (ff-EBMs).

- The bilevel optimization is solidly established with the inner problem accurately reflecting the equilibrium state search, crucial for EBMs, while the outer problem adeptly encapsulates the parameter adjustment to minimize the cost function. This nested structure is then expanded into a multi-level optimization problem that inherently captures the complexity of training ff-EBMs, where each level represents an optimization challenge within the model's architecture.

- The theoretical robustness is complemented by a well-thought-out algorithm that intertwines backpropagation and equilibrium propagation, offering a practical solution to the end-to-end training of these hybrid models. This algorithm is not just a theoretical construct; it is underpinned by a series of proofs that validate its effectiveness. The experimental results are particularly compelling, as they demonstrate the effectiveness of the proposed approach on ff-EBMs, achieving state-of-the-art performance on the ImageNet32 dataset.

**Weaknesses:**

While the paper introduces an innovative approach to hybrid digital-analog neural network training, there are a few areas where it shows some limitations.
- The paper could benefit from clearer articulation regarding the energy efficiency claims, providing more detailed comparisons with existing systems to substantiate these claims.
- Additionally, the reliance on simple datasets, ImageNet32, for experimental validation, while common, might limit the generalizability of the findings. A broader range of datasets with varying characteristics could strengthen the paper's conclusions. The choice of ImageNet32 should be justified with respect to its relevance to the research goals and its limitations for testing the model's robustness.
- Lastly, while the authors acknowledge the need for further research, a more explicit discussion on the current limitations and a roadmap for future work would provide a more comprehensive view of the research's trajectory.

**Questions:**

In the proposed implicit BP-EP chaining algorithm, is it necessary to ensure the satisfaction of the first-order optimality conditions derived from the implicit theory, or can the algorithm be robust and effective without strictly guaranteeing these conditions?

**Limitations:**

A notable limitation of our study is the constrained scope of network architectures and datasets utilized for validation. While the implicit BP-EP chaining algorithm has shown success with Deep Hopfield Networks on ImageNet32, its performance on the standard ImageNet and transformer-based models, which are more complex and widely used, is yet to be established.

---

> ### Author Rebuttal · Authors · 2024-08-06
>
> We thank Reviewer kDhg for their detailed review and we are happy that they've appreciated our work.
>
> i) *Energy efficiency claims, detailed comparisons with existing systems to substantiate these claims*. Please see our global rebuttal.
>
> ii) *Why only ImageNet32?*
>
> ImageNet32 was used for two reasons:
> a) we wanted to align our work with the most recent EP literature evaluating algorithms on vision tasks only with dimensionality at most 32 [1, 2, 3];
> b) The reason we restricted ourselves to such a low-dimensional dataset is because ff-EBM training simulations are already *relatively slow* on simple datasets – see below for a more detailed discussion.
>
> iii) *Explicit discussion on the current limitations and a roadmap for future work*.
>
> We herein detail the roadmap for future work which addresses three major limitations of our  paper: 1) Scalability to complex tasks; 2) Generalizability to other datasets and architectures; 3)  Hardware realistic training of ff-EBMs.
>
> **Please find the associated Fig. 3 inside the PDF attached to our global rebuttal**.
>
> 1) **Scalability to complex tasks**, i.e. *higher input dimensionality* (e.g. going from ImageNet32 to ImageNet224) and *deeper and larger architectures*. As we wrote to Reviewer k7RS, we insist that what bottlenecks the scalability of our work is not the proposed algorithm itself but the **ability to efficiently simulate ff-EBM inference**, because of the lengthy root finding algorithms needed to compute equilibria inside the EB blocks. As highlighted by Table 2, training a ff-EBM of 15 layers on ImageNet32 takes at least 40 hours, *regardless of the training algorithm used*. Therefore, “scaling” either requires:
>  - a) *maintaining the current architecture*, i.e. with Deep Hopfield Networks (DHNs) as EB blocks, on ImageNet224, *with more GPUs and more compute time*, or:
>  - b) *changing the EB block*, i.e. use EB blocks with better convergence properties than Deep Hopfield Networks (DHNs). A compelling class of models is called Deep Resistive Nets (DRNs) [4] which have the double advantage to be more hardware plausible and faster to simulate owing to their convex energy function.
>
> 2) **Generalizability to other datasets and architectures**. As pointed out by Reviewer 9JF8, our approach has yet to be proved on *transformer architectures* and associated datasets, which subsumes:
>
> - a) *Defining a ff-EBM counterpart of standard transformers*. To map a transformer into a ff-EBM architecture, we need to break it up into feedforward / digital and energy-based / analog blocks. *As illustrated in Fig. 2 inside the PDF attached to our rebuttal*, the ff-EBM counterpart of a transformer encoder block would comprise: i) a feedforward block with normalization and attention layers, ii) an energy-based block with the two fully connected layers. We denote below this ff-EBM counterpart of a transformer architecture “TF-ff-EBM”.
>
> - b) *Testing a TF-ff-EBM with DHN blocks*. As we did in this study, we would use *DHNs as EB blocks* in the proposed TF-ff-EBM architecture and test it on a relatively simple dataset to start with (e.g. the PTB dataset).
>
> - c) *Testing a TF-ff-EBM with DRN blocks*. See 3a) below.
>
> - d) *Scaling hardware-realistic training of TF-ff-EBM to more complex tasks*. We would scale the previous approach to deeper TF-ff-EBMs on the Wikitext datasets, with several additions to the architecture depicted in c) to make it more hardware realistic (see 3c) below).
>
> 3) **Hardware realistic training of ff-EBMs**. Moving towards more realistic proof-of-concepts of mixed-precision training of ff-EBMs, the following milestones need to be hit applying to both vision models considered in this paper as well as language models to be investigated:
>
> - a) *Using DRNs as EB blocks*.  As mentioned earlier, DRNs are *exact* models of nonlinear resistive analog circuits, made up of resistors, diodes, operational amplifiers with actual voltage nodes as neurons.
>
> - b) *Using quantized activations and gradients within feedforward blocks*. A realistic assumption for feedforward transformation would be to employ ***quantized** activations, weights and **gradients***. Techniques such as Quantization-aware Scaling (QAS) could be employed to compute meaningful quantized gradients inside feedforward blocks [5].
>
> - c) *Taking into account analog non-idealities*. As pointed out by Reviewer pevw, we need to consider a “detailed analog computing device model”, for instance taking “device variation” into account or any other device imperfection which may affect gradient computation. Taking these into account is crucial to assess how much extra digital circuitry it would take to mitigate these as it entails energy consumption.
>
> - d) *A comprehensive energy consumption analysis of hardware realistic ff-EBM training*. Finally, *quantitatively* and *precisely* assessing the energy efficiency gains of ff-EBM training with the more hardware-realistic assumptions introduced above is crucial.
>
> **We propose to add this detailed research roadmap inside the supplementary material of the camera-ready version of our paper in case of acceptance**
>
> iv) *Should the first-order optimality conditions be strictly guaranteed for the BP-EP chaining to work?*
>
> In practice, meeting optimality conditions can be quantitatively assessed with the static gradient analysis presented inside Fig. 3 and Fig. 4 (since $\lambda_\star$ can be typically be computed by automatic differentiation) and we observe that **the “more optimal” $s_{\star}^{\pm \beta}$ are** (in the sense of satisfying the KKT conditions) **,  the better the resulting training performance by EP**. For a further mathematical analysis of approximate first-order optimality in implicit models, we refer to [6, 7].
>
> [1] *Laborieux & Zenke, NeurIPS 2022*
> [2] *Laborieux & Zenke, ICLR 2024*
> [3] *Scellier et al, NeurIPS 2023*
> [4] *Scellier, ICML 2024*
> [5] *Lin et al, NeurIPS 2022*
> [6] *Zucchet & Sacramento, 2022*
> [7] *Meulemans et al, NeurIPS 2022*

---

> > ### Comment · Reviewer_kDhg · 2024-08-13
> > **Respond to rebuttal.**
> >
> > Thank you for your reply. The reviewer has no additional questions.

---

### Official Review · Reviewer_9JF8 · 2024-07-12

**Soundness:** 3
**Presentation:** 2
**Contribution:** 3
**Rating:** 7
**Confidence:** 4

**Summary:**

This paper proposes a new building model block with analog forward circuits and energy-based blocks built on a digital-analog hybrid setup.  A novel algorithm is further proposed to train the new block model. Experiments show the SOTA accuracy in the EP literature.

**Strengths:**

* This paper achieves SOTA accuracy compared to recent literature.
* A solid optimization method is proposed for the hybrid block.

**Weaknesses:**

* The paper's motivation is vague to me, as I am new to this topic. Why must we combine an analog forward circuit with a digital one for EP-based training? For the analog accelerator, we can use several other methods to do training, e.g., forward-forward only and zeroth-order optimization. What are the benefits of incorporating an energy-based model?

**Questions:**

N/A

---

> ### Author Rebuttal · Authors · 2024-08-06
>
> We thank Reviewer 9JF8 for their comments and we are happy that they've appreciated our work.
>
> i) *Why must we combine an analog forward circuit with a digital one for EP-based training?*
>
> We first want to clarify that per our modelling choice, analog and digital parts are respectively modeled as energy-based and feedforward models. Therefore, we do *not* assume that feedforward models are mapped onto analog. Modelling analog parts as energy-based models is a plausible assumption as it was shown that arrays of analog resistive devices minimize an energy function at equilibrium called the “pseudo-power” function [1, 2]. We kindly invite Reviewer 9JF8 to read details about these aspects inside the global rebuttal.
>
> Additionally, EP-based training does not  *by itself* require the combination of analog (i.e. energy-based) and digital (i.e. feedforward) circuits. EP only applies, by default, to models which are *fully* energy-based, and that would map to a *fully* analog system. Rather, this combination is a realistic hardware design constraint, as analog hardware can currently only easily support vector-matrix multiplications and certain non-linear functions, with all other operations (e.g. max pooling, batchnorm) carried out in digital (see Fig. 1 of the PDF). Therefore, as such a system may not be *fully* energy-based but only *by parts*, it was unclear, before this work **how EP training would extend to such systems**.
>
> Lastly, there is also an algorithmic advantage to use ff-EBMs as abstraction of such systems over EBMs: they are **faster to simulate**. Please read Reviewer k7Rs's rebuttal for further details, along with new results inside Table 1 of the PDF attached to our global rebuttal.
>
> ii) *What are the benefits of incorporating an energy-based model, compared to using forward-forward only or zeroth-order optimization?*
>
> This is an excellent question. Assuming Reviewer 9JF8 refers to the “Forward-Forward” (FF) algorithm [3] when mentioning “forward-forward only” optimization, both FF and zeroth-order (ZO) techniques apply to *any* model, be it energy-based or not, and estimate gradients through multiple forward passes through the *same* circuit, which is the holy grail for analog substrates. So why considering energy-based models at all?
>
> FF and ZO algorithms, while being mechanistically appealing for analog hardware [4], do not match the performance of automatic differentiation (i.e. backprop for feedforward models) on *equal* models, of around the same size as those studied in our work.
>
> - **ZO**. When using random weight perturbation [5] (WP, the best-known variant of ZO optimization, also known as “SPSA” [6]), resulting gradient estimates, although unbiased, have variance which scale cubically with the model dimensionality [7], yielding a significant gap in resulting model performance compared to backprop. This issue can be mitigated by architectural changes allowing the use of layer-wise, or even patch-wise losses [7,8], effectively reducing dimensionality and therefore the variance of ZO gradients or using small auxiliary networks to compute good “guesses” in the weight space (instead of random ones) [8]. These heuristics, while being effective at nearing backprop performance when training feedforward models from scratch, still don’t entirely close this performance gap on equal models. For instance, training a ResNet-18 by weight perturbation yields in the best case 45.8% top-1 test accuracy on ImageNet32 (using in this case auxiliary networks) while BP achieves 53.7% on the same architecture [8]. In contrast with ZO optimization, the use of EP on EB models or ff-EBMs **always match the performance of automatic differentiation on equal models**, in a principled fashion and without any heuristic, as observed in this work and past EP works [9, 10, 11]. Indeed, the energy-based requirement guarantees that EP gradients match automatic differentiation gradient in the limit of small nudging [12, 13].
> This being said, we are aware that ZO techniques may however be sufficient to *fine-tune* pre-trained models [14], as pre-trained models somehow behave as small models (i.e. their Hessian has small rank). Therefore, neither first-order optimization techniques as EP, nor energy-based models, may be necessarily needed in the fine-tuning context, depending on the difficulty of the downstream task.
>
> - **FF**. The FF algorithm is endowed with the same high-level features of WP or ZO order algorithms, namely with the use of multiple forward passes with the same circuit and of local losses. However, it is a heuristic algorithm with no theoretical guarantees: the weight update does not estimate the gradient of the loss of interest, nor is it guaranteed to decrease the loss. Instead, the algorithm is designed to increase (resp. decrease) the “goodness” of each layer on positive (resp. negative) samples, with the goodness function and negative samples being heuristically motivated [3]. As a result, the resulting model accuracy trained by FF on CIFAR-10 does not exceed 55% on CIFAR-10 [15].
>
> In summary, both ZO and FF algorithms, while possessing certain qualities and advantages similar to EP, are as yet not known to scale comparably to BP, whereas our algorithm (in principle) does.
>
>
> **We propose to add this detailed discussion around FF and ZO in the related work section**
>
> [1] *Johnson, 2010*
> [2] *Kendall et al, 2020*
> [3] *Hinton, 2022*
> [4] *Oguz et al, 2023*
> [5] *Fiete et al, 2010*
> [6] *Spall, 1992*
> [7] *Ren et al, 2022*
> [8] *Fournier et al, 2023*
> [9] *Laborieux et al, 2021*
> [10] *Laborieux & Zenke 2022*
> [11] *Scellier et al 2023*
> [12] *Ernoult et al 2019*
> [13] *Scellier & Bengio 2019*
> [14] *Malladi et al, 2023*
> [15] *Aminifar, 2024*

---

> > ### Comment · Reviewer_9JF8 · 2024-08-12
> > **Thank you for your rebuttal**
> >
> > Thank you for your rebuttal.
> >
> > First, thank you for your new Figure 2, which makes your problem statement much clearer to me, while your previous introduction is too complex and hard to get your point. So, if I understand correctly, **you propose to make the analog part to an energy model and set the digital part to the normal feedforward layer. Then, you derive the automatic differentiation method to train this hybrid setup**.
> >
> > So, I have several key follow-up questions to help me decide whether to raise/lower my score.
> > * Which is the main source of the accuracy improvement? The hybrid setup is where the digital feedforward improves accuracy, or the beyond zeroth-order backpropagation algorithm is used.
> > * The reason I ask ZO/FF is not due to the accuracy; it is because ZO/FF is hardware-friendly for analog circuits where you cannot use backpropagation to do on-chip training. Therefore, ZO/FF is practical for analog hardware requiring forward passes. Is your proposed method also friendly? How can you obtain the right gradient through the analog circuit? Please discuss the real hardware implementation for your method, especially how to obtain the gradient.
> >
> > thank you so much.

---

> > > ### Author Response · Authors · 2024-08-12
> > > **Source of improvement & hardware implementation of our algorithm**
> > >
> > > We are glad that Reviewer 9JF8 appreciated our rebuttal, and are indeed happy to clarify their questions.
> > >
> > > - *you propose to make the analog part to an energy model and set the digital part to the normal feedforward layer*. Yes: analog and digital parts account for energy-based and feedforward models respectively.
> > >
> > > - *you derive the automatic differentiation method to train this hybrid setup*. To be certain to be on the same page as Reviewer 9JF8, we slightly reformulate this sentence:
> > >   + we apply the *Lagrangian method* to derive optimality conditions, i.e. “KKT” conditions, to “train this hybrid setup” (Appendix A.2).
> > >   + the application of this method yields an algorithm which is itself a **hybrid** differentiation method which uses standard backprop (i.e. “automatic differentiation”) inside feedforward / digital parts and **equilibrium propagation inside energy-based / analog parts**.
> > >
> > > Therefore, our algorithm isn’t “pure” automatic differentiation end-to-end, but *only within feedforward parts of the model*.
> > >
> > > - *Which is the main source of the accuracy improvement?* The main source of accuracy improvement, *with respect to the past EP works*, is simply the *depth* of the model trained: the ff-EBMs trained here are twice as deep as the largest EBMs trained in previous EP works. Reviewer 9JF8’s question may then translate to: why can you train deeper networks? The most important reason, which is highlighted in Table 1 of our PDF and in our rebuttal to k7Rs, is that **ff-EBMs are faster to simulate than their fully EBM counterparts** (up to 4 times faster per Table 1). Instead of the superlinear scaling of the convergence time of DHNs (with respect to the number of layers) empirically observed in past EP works, the convergence time ff-EBMs made up of DHNs are EB blocks are guaranteed, by construction, to scale linearly with the number of blocks, the convergence time of a single block decreasing with its size. On the algorithmic side: since i) our algorithm performs on par with end-to-end automatic differentiation on the same ff-EBMs on the one hand, and that ii) ZO techniques are expected to perform less well than automatic differentiation on equivalent architectures (as highlighted in our rebuttal to Reviewer 9JF8), we can conjecture that, on *equal ff-EBMs*, our algorithm may likely perform better than ZO *when applied end-to-end*.
> > >
> > > - *“Is your proposed method hardware-friendly? Please discuss the hardware implementation of your method”*. First, as highlighted in our rebuttal, our method extends EP-based training to scaled hardware systems which: i) may not fit a single analog core, ii) may still require operations which cannot be supported on analog hardware and may instead require digital, high precision hardware. Therefore, Fig. 1 of the PDF attached to our rebuttal is a plausible depiction of the hardware implementation of our method and taking these constraints into account, our algorithm is **hardware plausible**. Second, we insist that EP-based training inside each of the analog / energy-based block inherits the “hardware-friendly” / FF-like features of EP training: in our algorithm, **gradients inside analog / energy-based blocks are computed using only forward passes / relaxations to equilibrium**. A plausible hardware implementation of this analog blocks are *deep resistive networks* [1, 2]. Lastly, feedforward blocks, which are maintained in digital, could also directly leverage quantization algorithms [3] and even **ZO algorithms** (as mentioned in L.290-291 of our paper) to facilitate their implementation on memory-constrained, low-power hardware.
> > >
> > > To summarize: our algorithm makes the best of analog and digital worlds, by: i) being hardware plausible at the system level (Fig .1 of PDF), ii) preserving the “hardware-friendliness” of FF-like learning inside EB/analog blocks, iii) possessing the ability to leverage any quantization algorithm inside feedforward blocks and iv) possessing the ability **to apply ZO algorithms** instead of backprop (as currently done with our algorithm) **inside feedforward blocks**.
> > >
> > > Finally, we would like to thank reviewer 9JF8 for drawing our attention to some critical points where our paper could better communicate some of the high-level motivations and details of our algorithm. In addition to what was proposed in our original rebuttal, we would like to propose adding a pseudo algorithm in the appendix showing how ZO could be applied within feedforward blocks (instead of backprop currently) such that gradients would be computed **everywhere with forward passes only**, i.e. both inside analog blocks (by EP) and feedforward blocks (by ZO) (as first suggested in L. 290-291 of our paper).
> > >
> > > [1] Kendall et al (2020). Training end-to-end analog neural networks with equilibrium propagation.
> > >
> > > [2] Scellier, B. (2024). A Fast Algorithm to Simulate Nonlinear Resistive Networks. ICML 2024.
> > >
> > > [3] Lin et al (2022). On-device training under 256kb memory. NeurIPS 2022

---

> > > > ### Comment · Reviewer_9JF8 · 2024-08-13
> > > >
> > > > Thank you so much for your rebuttal and for successfully addressing my concerns.
> > > > I am happy to raise my rating!

---

> > > > > ### Author Response · Authors · 2024-08-13
> > > > >
> > > > > Dear Reviewer 9JF8, thank you so much for taking the time to engage in a discussion with us and for raising our score! Your remarks will greatly help improve our manuscript for readers interested in ZO optimization. Thank you again!

---

### Official Review · Reviewer_k7Rs · 2024-07-15

**Soundness:** 3
**Presentation:** 3
**Contribution:** 3
**Rating:** 5
**Confidence:** 3

**Summary:**

This paper presents Feedforward-tied Energy-based Models (ff-EBMs), a hybrid model that integrates feedforward and energy-based components, accounting for both digital and analog circuits. A novel algorithm is proposed to compute gradients end-to-end in ff-EBMs by backpropagating and "eq-propagating" through feedforward and energy-based sections, respectively, allowing EP to be applied to more flexible and realistic architectures. It has been shown that ff-EBMs can be trained on ImageNet32, achieving new state-of-the-art performance in the EP literature with a top-1 accuracy of 46%.

**Strengths:**

-- The proposed ff-EBMs as high-level models of mixed precision systems, where the inference pathway is composed of feedforward and EB modules, is interesting and novel. Specially, gradients computations as an end-to-end backpropagation through feedforward blocks and “eq-propagating” through EB blocks.

-- The results are also encouraging specially on CIFAR datasets.

-- The paper is easy to read and understand.

**Weaknesses:**

-- The primary limitation of this work is its accuracy performance on the ImageNet dataset. Although the paper has set a new state-of-the-art accuracy, it still lags behind the results achieved by Transformers and CNNs. There is a lack of compelling reasons to use this method given its comparatively lower performance.

-- Additionally, it is crucial to measure the energy consumption of this training method and compare it with traditional methods. How much energy savings does your method offer?

**Questions:**

See weaknesses.

---

> ### Author Rebuttal · Authors · 2024-08-06
>
> We thank Reviewer k7Rs for their comments and we are pleased they've appreciated our work.
>
> i) *Accuracy performance on the ImageNet dataset*
>
> As  acknowledged inside the “Limitations and future work” paragraph (L.302-305), ff-EBM training by EP remains to be proved at scale on deeper models and more complex tasks (see Fig. 3 of the PDF about our research roadmap). Therefore, we  fully agree that the resulting performance of ff-EBMs **trained by EP** on ImageNet “lags behind” SOTA performance achieved by modern deep learning architectures, such as transformers or deeper CNN models, **trained by BP** on the same dataset.
>
> However, when comparing *different* models (ff-EBMs and transformers) trained by *different* training algorithms (EP and BP respectively), it is hard to disentangle improvements that come from the model *or* the training algorithm taken separately. Our results displayed inside Tables 1 and 2 of our original submission clearly show that **EP always performs as well as the automatic differentiation baseline on equal models**. This suggests (we believe) that the scalability of our approach is not limited by the proposed algorithm *itself*, but by the **ability to efficiently simulate large ff-EBMs**. To further support this, note that (as seen in Table 2)  training a ff-EBM of 15 layers takes at least 40 hours, *regardless of the training algorithm used*. These simulation times come from the lengthy fixed-point iteration used to compute equilibrium states inside EB blocks.
>
> Therefore, we envision two paths towards achieving better performance on ImageNet (see Reviewer kDhg's rebuttal and Fig. 3 of our PDF for further details):
> - using more GPU resources for longer times on deeper ff-EBMs, whilst preserving the core architectural components of the ff-EBM used in the present paper;
> - using EB blocks which converge faster to equilibrium. As mentioned in L. 297, using Deep Resistive Networks [1] instead of Deep Hopfield Networks as EB blocks, or even employing techniques such as Anderson Acceleration for root-finding, would yield a significant speed-up which would allow to train much deeper ff-EBMs.
>
> **We propose to make it clearer, in the “Limitations and future work” paragraph, that our simulations are bottlenecked by the ability to simulate large-scale ff-EBMs rather than the proposed algorithm itself, and will accordingly mention the two research paths mentioned above**
>
> ii) *There is a lack of compelling reasons to use this method given its comparatively lower performance*
>
> - *Clarification*. We would like to clarify that the goal of this work is only to bring a **proof-of-concept** to guide the design of mixed-precision training accelerators with analog and digital parts. As ff-EBMs remain inefficient to simulate on GPUs, we do not advocate any practical use of ff-EBM over standard modern feedforward architectures, nor training them by EP instead of automatic differentiation, to achieve the best possible performance **when using GPUs**. For lack of widely accessible *real* mixed precision architectures to test our training algorithm against, ff-EBMs are high-level abstractions thereof which we *simulate* using GPUs.
>
> - ***ff-EBMs are much faster to simulate than their EBM counterparts***. With regards to **simulating** hardware systems as mentioned above, a compelling reason which did not appear clearly enough in our original submission is that **ff-EBMs are much faster to simulate than their single-block EBM counterpart of same depth**. To demonstrate this, we re-ran experiments of Table 1 (the "splitting experiment", section 4.3) with a *slightly different method to compute equilibria inside EB blocks*. We contrast these two settings below:
>   + *Former setting of the original submission* (Table 1 of the paper). Our initial goal was to demonstrate that starting from a standard ("single-blocked") EBM that ff-EBMs of *same depth* retained the same expressive power across all possible even splits, per the resulting performance on CIFAR-10. In this case, a fixed-point iteration scheme was applied inside each EB block, with the same *fixed* (and possibly large) number of iterations across all splits and throughout training. However, EB blocks of smaller size are expected to converge faster and therefore may require fewer fixed-point iterations [2-4].
>   + *New setting* (Table 1 of the PDF attached to the rebuttal). In order to take advantage of the faster convergence of smaller EB blocks and ensure the fairest comparison of wall-clock times across all splits *for a given depth* ($L=$6 or 12), we employ a *relative convergence criterion* to compute equilibria inside EB blocks. Namely, denoting $s(t)$ the state of a given EB block at the t-th fixed-point iteration, the fixed-point dynamics (Eq. 13 of our submission) are executed until $|(s(t + 1) - s(t))/ s(t)| < \epsilon$ where $\epsilon$ denotes some threshold. In spite of its simplicity, this trick was never employed in past EP works. Using this criterion, we can observe from Table 1 of the PDF attached to the rebuttal that ff-EBM training **can be up to 4 times faster that with equivalent EBM, while maintaining and often improving performance**.
>
> To summarize: instead of the **superlinear** scaling of the convergence time of DHNs (with respect to the number of layers) empirically observed in past EP works [2--6], the convergence time ff-EBMs made up of DHNs are EB blocks are guaranteed, *by construction*, to scale **linearly** with the number of blocks, the convergence time of a single block decreasing with its size.
>
> iii) *How much energy savings does your method offer*?
>
> We kindly invite the Reviewer to read our global rebuttal for this question.
>
> [1] Scellier, ICML 2024
> [2] Ernoult et al, NeurIPS 2019
> [3] Laborieux et al, Frontiers in Neuro., 2021
> [4] Laborieux & Zenke, NeurIPS 2022
> [5] Scellier et al, NeurIPS 2023
> [6] Laborieux & Zenke, ICLR 2024

---

### Official Review · Reviewer_Jt5z · 2024-07-31

**Soundness:** 3
**Presentation:** 1
**Contribution:** 3
**Rating:** 5
**Confidence:** 3

**Summary:**

Analog in-memory computing is gaining traction as an energy-efficient platform for deep learning. However, fully analog-based accelerators are challenging to construct, necessitating a training solution for digital-analog hybrid accelerators. This paper introduces Feedforward-tied Energy-based Models (ff-EBMs), a hybrid model that integrates feedforward components, typical in digital systems, with energy-based blocks suitable for analog circuits. An algorithm is derived to compute gradients end-to-end for training ff-EBMs. Experimental results show that ff-EBMs achieve superior accuracy on ImageNet32 classification compared to conventional equilibrium propagation literature.

**Strengths:**

1. This paper developed training algorithms specifically for digital-analog hybrid computing platforms, addressing a realistic computing scenario that leverages analog in-memory computing.

**Weaknesses:**

1. The paper lacks examples of situations where ff-EBM is needed. It does not adequately explain which parts of the overall system are digital and which are analog when using ff-EBM. Additionally, it does not clarify whether the existing training techniques can be used in these scenarios or what advantages ff-EBM offers compared to traditional methods.

**Questions:**

As I understand it, analog in-memory computing naturally finds the node voltage that minimizes the energy function through Kirchhoff's laws, classifying it as an EP problem. Therefore, if we disregard the natural minimization of the energy function, the training process with analog in-memory computing appears very similar to the conventional backpropagation-based training procedure. Hence, the derived algorithm for calculating end-to-end gradients of the ff-EBM seems trivial. In this context, I don't see the difference between this research and previous studies that have conducted training on analog in-memory systems using EP or backpropagation. Could you explain the differences between the previous approaches and the proposed approach in more detail? Providing examples of situations where ff-EBM is needed, as discussed in the Weakness section of this review, would be helpful.

**Limitations:**

Please check the weakness.

---

> ### Author Rebuttal · Authors · 2024-08-06
>
> We thank Reviewer Jt5z for their honest feedback. We consider these clarifications important to highlight our contribution for readers.
>
> i) *Difference between this research and previous studies that have conducted training on analog in-memory systems using EP or backpropagation?*
>
> We detail two research trends and contrast them with our work:
>
> a) **Training analog systems using BP**. In this context, analog hardware sustains **feedforward models**, with two main techniques:
>
> - *On-chip BP training*: BP is directly executed on the analog hardware itself alongside inference. In general, forward and backward passes are necessarily executed on two **distinct** dedicated circuits. However, weights and activations must be *transported* from the inference circuit to the gradient computation circuit. Since weights/activations are encoded as noisy analog quantities (upon read and write operations), transporting them yields a *device mismatch* between these two circuits affecting in turn the resulting training performance [1].
>
> - *Off-chip BP training*: a digital proxy of the analog system is trained by BP on GPUs and the resulting weights are mapped onto the analog system, yielding in turn an analog *inference-only engine* [2]. This approach circumvents the difficulty of on-chip training of analog systems.
>
> b) **Training analog systems using EP**. When considering **energy-based** (EB) instead of feedforward models, the *same circuit* can be used for both the forward and backward passes. Therefore, when mapping an EB model onto a *single* analog circuit, the resulting system no longer suffers from device mismatch [1], with all the quantities required to compute gradients locally available at the same location, unlike BP. See global rebuttal and PDF for greater details.
>
> c) **This work**. While it is clear how to map models *small enough to fit single analog chip* and with standard weight stationary operations such as fully connected layers onto energy-based, analog compute engines trainable by EP [3, 4], models spanning *multiple analog chips* alongside many feedforward operations (e.g. batchnorm) to be maintained in digital with high precision result in systems which may not be readily trainable by EP – see Fig 1 of our attached PDF. **No existing work has tackled the extension of EP to this setting** where EP may only be applied within some subparts of the model, with BP applied elsewhere, in a principled fashion.
>
> **We accordingly propose to add an extra related work section related to this literature**.
>
> ii) *Examples of situations where ff-EBMs are needed?*
>
> See our global rebuttal and attached PDF. Additionally, the architecture considered for our experiments (L.213) is a very concrete example of this where it is unclear how to map batchnorm onto an energy-based, analog piece of hardware, and which may therefore be maintained in digital. Another example (mentioned L.304) would be transformer architectures – see Reviewer’s kDhg rebuttal and Fig. 2 of the PDF.
>
> iii) *Which parts of the overall system are digital and which are analog when using ff-EBM?*
>
> As explained in L.71-72, we “model digital and analog parts as feedforward and EB modules respectively”. This is further illustrated on Fig. 2, with “yellow and pink blocks denoting EB and feedforward transformations” and using the exact same notations, Eq. 12 explicitly defines the energy function of EB blocks and the feedforward transformations used for our experiments.
>
> **We propose to extend the color code used inside Fig. 2 to algorithms and equations to better emphasize which parts of the model and of the algorithm would be sustained in digital and analog**
>
> iv) *Can existing training techniques can be used in these scenarios and what advantages ff-EBM offer compared to traditional methods?*
>
> - As explained in L. 229, ff-EBMs can be trained by traditional automatic differentiation (AD) through the root finding algorithm used to compute the equilibrium state of the ff-EBM (Eq. 13), which here reduces to Implicit Differentation (ID) – its implementation is detailed in Algorithm 12 below L. 605 inside the Appendix.
>
> - About the advantages of **EP-based training of ff-EBMs** compared to ID or AD: we want to emphasize that **AD is only used here for simulation benchmarking purposes** to demonstrate that our algorithm achieves the best possible performance and would not be practical to deploy onto analog hardware for all reasons mentioned above. Namely, exactly as for BP as described previously, ID would require two separated circuits to compute *the steady state of the ff-EBM* on the one hand, and *the associated Lagrangian multipliers* on the other hand.
>
> - Another advantage of **ff-EBMs alone**: they are *faster to simulate than standard EBM counterparts* (see Table 1 of the PDF and details inside Reviewer's).
>
> **We propose to add a more detailed mathematical definition of AD and ID when applied to ff-EBMs to better emphasize the benefits of EP-based training of ff-EBMs**
>
> v) *Disregarding energy minimization, the proposed algorithm appears very similar to BP and therefore seems trivial*
>
> We assume that Reviewer Jt5z refers to the edge case where each EB block consists of a single layer such that the resulting ff-EBM is purely feedforward (“Recovering a feedforward net”, L.160), with our proposed algorithm reducing BP in this case (“Recovering backprop”, L.198, corollary A.5.1, Alg. 3 and Alg. 5 in appendix A.2). If so, we agree with Reviewer Jt5z **on this particular edge case**, and this is why **we discarded it upfront for our experiments** (L.201). However, in any other situation where EB blocks don’t reduce to a single layer, the proposed algorithm, deeply rooted in a Lagrangian-based approach thoroughly described in Appendix A.2, **doesn’t trivially reduce to any known algorithm in the literature**.
>
> [1] Yi et al, Nature Electronics, 2022
> [2] Wright et al, Nature, 2022
> [3] Kendall et al, ArXiV, 2020
> [4] Scellier, ICML 2024

---

> ### Comment · Reviewer_Jt5z · 2024-08-08
> **Additional questions on the novelty of this paper.**
>
> Thank you very much for your careful and detailed response.
>
> Figure 1 and Figure 2 in the attached PDF file clearly illustrate the hardware set-up and the proposed layer design. I acknowledge that this paper is pioneering in addressing the training algorithm for hardware composed of multiple analog chips, and I agree that this is a realistic configuration.
>
> However, I still have a few questions regarding the novelty of this paper. The ff-EBM model architecture comprises a chain of the FF module and the EB module. Given that the training algorithms for the FF module and EB module are established conventions (backpropagation and equilibrium propagation, respectively), I find it challenging to identify the difficulty in designing a training algorithm for ff-EBM. Since the training algorithms for the FF and EB modules are predetermined, the primary task for ff-EBM is to ensure the proper passing of gradients between the FF and EB modules. Nonetheless, passing gradients between these modules appears straightforward by following the chain rule.
>
> Therefore, I would like the authors to highlight any algorithmic challenges related to configuring the FF and EB modules within a single network architecture. More specifically, I would appreciate it if the authors could provide more detailed explanations on the challenges involved in passing gradients between the FF and EB modules and how these challenges are addressed.

---

> ### Author Response · Authors · 2024-08-08
> **Answering additional questions on the novelty of the paper**
>
> Dear Reviewer Jt5z,
>
> We thank you very much for engaging promptly in this discussion, and are happy indeed to read that the PDF file brought some satisfying clarifications about the relevance of the problem tackled.
>
> If we understand you correctly, since BP and EP are well established algorithms in their own right, chaining them inside a given architecture appears straightforward “by following the chain rule”. In fact, we are very pleased that you have drawn attention to this issue, as we are increasingly recognizing that some of the terminology we use could prove misleading, and to some extent confusing for readers. Therefore, as requested, we highlight below the specific “algorithmic challenges” pertaining to this chaining, “how [they] are addressed” and more broadly the “novelty of this paper”.
>
> - **EP-BP chaining is intuitive, but not trivial to derive**. You write that chaining EP and BP inside a given architecture “appears straightforward by following the chain rule”. We would like to focus attention on this question: what is meant by “following the chain-rule” in the context of ff-EBMs?
>
>   + Traditionally, “chain-rule” refers to gradient chaining inside *feedforward models*. Namely, let us assume a computational graph of the form $s^1 = F(x) \to s^2 = G(s^1)$. Assuming an error signal $\delta^2 = \partial_{s^2} L$, then the “chain rule” prescribes that the error signal at $x$ reads $\partial_{x}L = \partial_x F(x)^\top \cdot \partial_{s^1}G(s^1)^\top \cdot \delta^2$.
>
>   + Now let us assume instead the following “hybrid” computational path: $s^1 = F(x) \to s^2: \nabla_{s^2}E(s^2, s^1) = 0$. Given that $s^2$ in this case is an *implicit* function of $s^1$, i.e. there is no explicit G mapping as before between $s^1$ and $s^2$, how would one directly apply the above “chain-rule” here? To put it differently, how do we rigorously route error signals backward through this computational graph? This is the first “challenge” we addressed.
>
>   + Hence the need to derive gradient chaining inside ff-EBMs *rigorously, from first principles* by: 1) stating the learning problem as a multilevel constrained optimization problem (Eq. 8), 2) writing the associated Lagrangian (Eq. 21), 3) solving for the associated KKT conditions (Eqs. 22-34) for the primal variables (i.e. the steady states of the blocks) and associated Lagrangian multipliers (i.e. the error signals inside blocks). Therefore, this is how “we addressed” the above challenge.
>
> We emphasize that **this derivation** (namely Theorem 3.1 about the rigorous chaining of EP and BP gradients) and the **resulting explicit and implicit chaining algorithms** (Alg. 4 inside Appendix A.3, Alg. 2 in the main) **are novel**. Also, note from the above that our algorithm comes into an *explicit* and *implicit* variant (see paragraph “Proposed algorithm: implicit BP-EP chaining” L.187 and Lemma A.4), the latter appearing as a “pure” EP implementation. As such, the fact that EP-BP chaining can be cast into a *non-trivial generalization of EP*, as appearing in Alg. 2 and Lemma A.4, is not self-evident neither.
>
> - **The experimental demonstration of this algorithm is also novel**. Finally, to further highlight the “novelty of this paper”, **our algorithm was never tested in practice before this work** and having it succeed on ImageNet32 came with lots of “challenges” as well, the most important one being the simulation time. We addressed this problem in two ways:
>
>   + As explained in the global rebuttal and in greater details inside Reviewer k7Rs’s rebuttal, splitting an EBM into several EBM blocks tied by feedforward modules results in an architecture that is not only more hardware realistic, but also **easier to simulate**. Indeed, instead of the superlinear scaling of the simulation time with respect to the number of layers observed in past EP works, our approach guarantees, by construction, that this **simulation time scales linearly with the number of blocks**, each of these blocks converging much faster than the full EBM counterpart. See our new table of results inside our PDF attached to this rebuttal. While the ff-EBMs trained are still relatively shallow, **they are twice as deeper as the deepest EBM trained by EP** in most recent related works.
>
>    + Finally, using *Gaussian Orthogonal Ensembles* (GOE) to initialize weights inside EB blocks was instrumental in having ff-EBM training experiments work.
>
> Finally, **we would like to propose the addition of a few sentences to our introduction highlighting this novelty**, and particularly the degree to which our algorithm is derived by exploiting an intimate theoretical connection between energy based learning and **implicit differentiation**, rather than literal "backprop" as applied in standard feedforward nets where the standard "chain rule" applies. Hence our claim that this contribution belongs to the realm of EP, and *implicit learning* more broadly. Again, we thank you for drawing our attention to this.

---

> ### Comment · Reviewer_Jt5z · 2024-08-11
>
> Thank you for your detailed explanation.
>
> Your response has helped me better understand the proposed work, and I have made every effort to assess its value.
>
> Firstly, I am increasing my score to 5, as the novelty of your work is now clear to me. From my understanding, EP is designed to be applicable to any network architecture, including those with feedforward blocks, as the minimization of energy can be aligned with the minimization of the objective function (as shown in Figure 1 and Chapter 3 of the BP paper [1]). Therefore, I still view Eq. (8) as a straightforward integration of FF and EBM. However, I acknowledge that the gradient calculation starting from Eq. (8) is not trivial, and the detailed derivation of these gradients is a key novelty of your work.
>
> While I believe this work is highly significant in the field of analog-based AI accelerator systems, I am hesitant to raise my score further because the presentation of your work could be improved to better highlight its true value.
>
> In my opinion, the design of networks for analog-based computing is heavily constrained by hardware considerations, as there are significant challenges in scaling fully analog systems. Analog computing is known for its energy efficiency compared to digital computing, while digital computing offers greater scalability. To build an efficient yet scalable AI acceleration system, a hybrid design is essential. In this context, I believe the true value of your work lies in extending the scalability of EP-based models for analog computing by integrating FF modules. For this reason, I think that when adopting ff-EBM, the key concern should be scalability. The scalability of ff-EBM, as compared to using EBM alone, is a critical point that should be thoroughly explored. For example, with EBM alone, only a single analog-based unit in Figure 1 of attached PDF could be used for a single network. Obviously, ff-EBM should be much better than EBM, but I believe this paper needs detailed discussion on the accuracy of this EBM-based model and how it compares to the accuracy of a larger model designed with ff-EBM that fully utilizes the entire system depicted in Figure 1 of attached PDF, as this is an important aspect of this work.
>
> [1] B. Scellier and Y. Bengio. Equilibrium propagation: Bridging the gap between energy-based models and backpropagation. Frontiers in computational neuroscience, 11:24, 2017.

---

> ### Author Response · Authors · 2024-08-12
> **Clarifying Scellier-Bengio's EP paper & scalability of ff-EBMs**
>
> We are very grateful to Reviewer Jt5z for spending time to understand the value of our paper, subsequently increasing our score and engaging in this discussion which will tremendously benefit the presentation of our work. Thank you so much!
>
> In the light of Reviewer’s Jt5z last answer, we would like to clarify some essential points they raised:
>
> -  *"EP is designed to be applicable to any network architecture, including those with feedforward blocks"*. As mentioned in Section 2.3 of our paper, **EP only applies to energy-based models**. Fig. 1 of the seminal EP paper [1] is indeed misleading: when writing “Equilibrium Propagation applies to any architecture”, one should understand “any architecture **topology** so long as it derives from an energy function”. Indeed, as indicated by the title of the section 3 of this paper, EP really is a “Machine Learning Framework **for Energy-Based models**”: their Figure 1 is only meant to emphasize that EP applies to *any* energy-based models, not necessarily *layered* energy-based models. In this context: **layered does not meant feedforward**. As Scellier & Bengio write themselves: “*In particular, the [EP learning rule] holds for any architecture and **not just a layered architecture** (Figure 1) like the one considered by Bengio and Fischer (2015)* [which is also an EB model]”.
>
> -  *"Therefore, I still view Eq. (8) as a straightforward integration of FF and EBM"*. Given the clarification above, this conclusion may no longer hold, especially when noticing that Eqs.17-18 of the seminal EP paper [1] (in the section 3 mentioned by Reviewer Jt5z), i.e. the bilevel program the EP algorithm solves, **is an explicit particular case of the Eq. 8 of our paper**, i.e. the multilevel program that our algorithm solves.
>
> -  *"I think that when adopting ff-EBM, the key concern should be scalability"*. We totally agree! Investigating the scalability of ff-EBM training by our algorithm on deeper ff-EBMs, more complex tasks and exploring new datasets and architectures is part of our research roadmap. See our detailed answer to Reviewer kDhg and associated Fig. 3 inside the PDF.
>
> - *"The scalability of ff-EBM, as compared to using EBM alone, is a critical point that should be thoroughly explored"*. We are happy Reviewer Jt5z mentions the importance of this comparison since our “splitting experiment” (Section 4.3) goes exactly in this direction. This experiment reveals that a single EBM block performs comparably to an ff-EBM **of equal depth** with various block sizes.
>
> [1] Scellier, B., & Bengio, Y. (2017). Equilibrium propagation: Bridging the gap between energy-based models and backpropagation. Frontiers in computational neuroscience, 11, 24.

---

### Author Rebuttal · Authors · 2024-08-06

We thank reviewers for their time and highly valuable comments. In light of these, we propose several clarifications which we hope render the value of our work more clear to readers and can quell some of the concerns expressed.

**I-Proposed additions to our paper**
- A  discussion about the **relevance of ff-EBMs** in analog computing and **energy efficiency gains of the proposed method** (see *inside this global rebuttal*), including a sketch of a hybrid chip with analog and digital parts with EP training at chip scale (Fig. 1 inside PDF).
- A **detailed roadmap for future research** (Reviewer kDhg’s rebuttal and Fig. 3 inside the PDF). This includes a  discussion about **hardware realism of ff-EBM training** (Reviewer pevw’s rebuttal).
- Explanation of **how transformer architectures can be mapped onto ff-EBM architectures** and hence trained by the proposed algorithm (Reviewer kDhg’s rebuttal and Fig. 2 of PDF).
- A  related work section about **training analog in-memory systems using EP or BP** (Reviewer Jt5z’s rebuttal).
- A related work section with **detailed comparison of energy-based learning with 0-th order optimization (ZO) and the forward-forward algorithm (FF)** (Reviewer 9JF8’s rebuttal).
- Detailed mathematical definition of automatic differentiation (AD) and implicit differentiation (ID) when applied to ff-EBMs **to better emphasize the benefits of EP-based training of ff-EBMs** (Reviewer Jt5z’s rebuttal).

**II- New experiments**

To further justify why ff-EBMs are useful in practice (Reviewers Jt5z, 9JF8), we highlight in Table 1 of the PDF that **they are much faster to simulate than EBM counterparts of equal depth** (see details below and inside Reviewer k7Rs's rebuttal).

**III-The relevance of ff-EBM modelling**

For some readers, “the paper’s motivation [can look] vague” (Reviewer 9JF8) and it may be unclear “when ff-EBMs [are] needed” (Reviewer Jt5z). Therefore, we clarify some fundamentals underpinning our work, alongside new results (Fig. 1 and Table 1 inside the PDF) of benefit to all Reviewers. Importantly, our answer below relies on an **end-to-end experimental realization of EP training on analog resistive networks** [6].

i) *Why analog computing?* Analog in-memory computing (AIMC) accelerators achieves greater energy efficiency by: a) encoding weights as conductance values of resistive elements in an analog crossbar array *which can be read and written at very low cost* [5]; b) leveraging analog physics, i.e. continuous physical observables, to perform *vector-matrix multiplications (VMMs) at very low cost* [1 – 5].

ii) *Why modelling AIMC systems as energy-based models?* Nonlinear resistive networks as AIMC systems are *energy-based* models [1, 4], as Kirchhoff laws, which govern these systems, obey a variational (**energy minimization**) principle. Energy-based models are convenient as they can compute loss gradients using multiple relaxations to equilibrium using a *single circuit*, as prescribed by EP. This is why, in line with these works, we also model AIMC cores as energy-based models.

iii) *Why are AIMC systems not sufficient alone?* As highlighted in the scalability study of an experimental realization of EP [6], **AIMC energy-based cores may only constitute the smallest compute unit of a larger system comprising many digital parts**. We adapt from [6] a simplified outline of an “architecture to implement [EP training] on a chip scale” in *Fig. 1 of the attached PDF*. Quoting from the authors of [6], “[this] architecture consists of **hybrid (analogue-digital) operations** [and is] hierarchical”: as seen from our Fig. 1, several analog processors form a tile, and several tiles form the chip, with digital buses connecting analog cores along with digital coprocessors to support analytical operations (e.g. maxpool, as in [6]). Therefore a **ff-EBM**, as inherently hierarchical and hybrid (Eq. 8 of our paper), **is a coarse-grained abstraction of such a modular architecture**, which EP is extended to in our work.

iv) **ff-EBMs are faster to simulate**. We reproduced the splitting experiments with a *uniform convergence criterion* to compute equilibria inside EB blocks across all splits (instead of tuning the number of iterations for each of these). **See Reviewer k7Rs's rebuttal for details and Table 1 of the PDF**.

**IV-Energy efficiency of ff-EBM training**

Reviewers k7RS, kDhg and pevw asked about the “energy consumption of [our] training method” and “detailed comparisons with existing systems”. As mentioned earlier, **energy savings fundamentally would come from lesser costs for VMMs and read/write operations on resistive devices inside EB blocks**. Providing an accurate answer to this question requires research beyond the scope of this paper (see Fig. 3 for a detailed outline of our research roadmap) and is highly system dependent. Based on [6], it is possible to convey this complexity and provide an estimation of the energy gains *compared to a NVIDIA V100 device* (for inference + gradient computation + weight update) depending on a variety of  factors. Most importantly:

- *System’s size*: if the model fits a single AIMC core (a 64x64 array of RRAM devices), energy gains are around $10^5$. If instead considering a model spanning multiple cores within the hybrid architecture previously described from [6], **which ff-EBMs most resemble**, energy gains reduce to $10^4$ because of the “architectural overheads, dominated by the peripheral circuitry” and “analog-to-digital conversions” [6]
- *Resistive devices*: the above numbers assume RRAM resistive devices. Considering Flash memory, writing is around 100x more energy consuming than on RRAM, yielding lesser energy savings if batch size is small. These numbers also change for transistor-based [2, 3] or magnetic-based [6] synapses.

[1] *Kendall et al, 2020*
[2] *Dillavou et al, 2022*
[3] *Dillavou et al, 2023*
[4] *Scellier, 2024*
[5] *Burr et al, 2017*
[6] *Yi et al, Nature Electronics, 2022*

---

### Decision · Program_Chairs · 2024-09-25

**Decision:**

Accept (spotlight)

**Comment:**

The authors present feedforward tied Energy Based Models (ff-EBMs) a new class of models that combine traditional feedforward layers with EBM layers. The paper also describes an end-to-end training methodology for ff-EBMs that chains the gradient estimations via Backprop (for ff layers) and Eq Prop (for EBM layers). This training methodology is supported by a detailed proof that confirms its correctness and shows that the ff-EBM training methodology works for any arbitrary split of a network into ff/EBM layers.

In the experiments section, the authors apply ff-EBM models for CIFAR10/CIFAR100/ImageNet32 datasets and show competitive accuracy. They also validate the equivalence of the gradients estimated using the proposed training methodology against traditional backprop based gradients and the validity of their training methodology for a range of ff/EBM splits for target model.

While the paper presents a strong theoretical explanation of ff-EBMs training and inference, many reviewers raised interesting questions about why ff-EBMs should be of interest and what are the benefits of Energy based models. The Authors have provided a detailed response to every question and provided useful figures and explanations of ff-EBMs usefulness and the HW system they have in mind to deploy these models.

A couple of points that stood out to me in terms of this work’s importance are as follows. Digital AIML HW implementations show high accuracy but suffer from high power cost, Analog HW implementation promises to counter some of this high-power cost. However, the analog HW suffers from reliability concerns and slow weight updates, they aren’t suitable for backprop based training and non-weight stationery operations (e.g. Batch Normalization). While EBMs offer a solution that enables reliable training on analog HW, it still depends on Digital HW for implementing non-weight stationery operations. The proposed ff-EBMs are a pioneering solution to combine the best of both worlds, by using analog HW wherever it enables low energy operations and outsourcing the rest to digital HW. Thanks to the training methodology demonstrated in this paper, any given model can be arbitrarily split onto analog/digital HW.

Further, EBMs currently suffer from long simulation times due to the delays involved in the root-finding step for each layer of EBMs. By combining ff-EBMs the authors speed up the training steps significantly (~4x as shown in Table 1 of rebuttal pdf). This in turn enables deploying networks with deeper EBM layers interspersed with ff layers to achieve superior accuracy compared to existing EBMs.

The authors also add a detailed timeline (Figure 3 rebuttal pdf) of their future work indicating extension to other datasets (ImageNet244) and model architectures (Transformer). In conclusion, ff-EBMs and their proposed training algorithm is a pioneering step in the direction of hybrid (digital/analog) HW systems implementing efficient AIML algorithms. This paper should be accepted for NeurIPS.

Edit: Updated my recommendation to spotlight (prev Oral) post discussion with SAC.